# Reconstructing winter climate anomalies in the Euro-Atlantic sector using circulation patterns

Erica Madonna[1], David S. Battisti[2], Camille Li[1], and Rachel H. White[3]

[1]Geophysical Institute, University of Bergen and Bjerknes Centre for Climate Research, Bergen, Norway
[2]Department of Atmospheric Sciences, University of Washington, Seattle, WA, USA
[3]Department of Earth, Ocean and Atmospheric Sciences, University of British Columbia

**Correspondence:** Erica Madonna (erica.madonna@uib.no)

**Abstract.** The efficacy of Euro-Atlantic circulation regimes for estimating wintertime climate anomalies (precipitation and surface temperature) over Europe is assessed. A comparison of seasonal climate reconstructions from two different regime frameworks (cluster analysis of the low-level zonal flow, and traditional blocking indices) is presented and contrasted with seasonal reconstructions using the North Atlantic Oscillation (NAO) index. The reconstructions are quantitatively evaluated using correlations and the coefficient of efficiency, accounting for misfit in phase and amplitude. The skill of the various classifications in reconstructing seasonal anomalies depends on the variable and region of interest. The jet and blocking regimes are found to capture more spatial structure in seasonal precipitation anomalies over Europe than the NAO, with the jet framework showing generally better skill relative to the blocking indices. The reconstructions of temperature anomalies have lower skill than those for precipitation, with the best results for temperature obtained by the NAO for high-latitude and by the blocking framework for southern Europe. All methods underestimate the magnitude of seasonal anomalies due to the large variability in precipitation and temperature within each classification pattern.

## 1 Introduction

Seasonal precipitation and temperature anomalies over Europe exhibit large year-to-year variability, with direct societal impacts such as on crop yields and renewable energy production (Grams et al., 2017; Jerez et al., 2013; Lesk et al., 2016). The seasonal climate signal results from a succession of daily weather that is often organized by the large-scale flow into a finite number of preferred circulation patterns, also called regimes (Corte-Real et al., 1995; Molteni et al., 1990; Vautard, 1990). Therefore, one might expect that seasonal climate anomalies could be reconstructed from the frequency of the dominant atmospheric patterns over a season. This approach has been used to study future trends in European precipitation (Santos et al., 2016) and as a predictor for droughts (Lavaysse et al., 2018). It can also be used to better understand and anticipate the changes in weather patterns that sum to the seasonal to decadal climate anomalies that stem from predictable changes in atmosphere-ocean

system, such as those associated with the El Niño-Southern Oscillation phenomenon and the Atlantic Multidecadal Variability associated with the ocean overturning circulation in the North Atlantic (see Battisti et al. (2019) for a review).

Atmospheric variability patterns can be characterized in several ways, such as by the North Atlantic Oscillation index (NAO, e.g. Hurrell, 1995), the occurrence of blocking (e.g. Pfahl and Wernli, 2012; Sillmann and Croci-Maspoli, 2009; Trigo et al., 2004), and the configuration of the North Atlantic jet stream (Woollings et al., 2010; Madonna et al., 2017). All these complementary classifications have been shown to be able to capture specific aspects of the wintertime climate over Europe, but there has been to our knowledge no direct comparison of the skill of these classifications in reconstructing seasonal climate.

In this study we compare how well European winter conditions are described by indices of the NAO, blocking, and North Atlantic jet configurations. The NAO manifests as sea level pressure (SLP) fluctuations with anticorrelated extrema between two poles, one over the Azores and one over Iceland. These fluctuations signify changes in the prevailing westerly winds and in the propagation path of storms into Europe (e.g., Hurrell, 1995; Hurrell et al., 2003; Qian et al., 2000; Rogers, 1997; Wanner et al., 2001). In contrast, blocking is the presence of a persistent and stationary high-pressure system that obstructs or deviates the westerly flow (Rex, 1950). In the North Atlantic sector blocking occurs mainly over three regions: Greenland, Scandinavia, and the Iberian Peninsula (Treidl et al., 1981; Davini et al., 2014, also shown in Figure 1). Precipitation is reduced within the blocked region (Sousa et al., 2017) while cold temperature extremes are often observed southeast of the blocked region (Sillmann and Croci-Maspoli, 2009). Lastly, jet configurations describe preferred flow paths of the North Atlantic jet stream, which acts as waveguide for midlatitude storms (Athanasiadis et al., 2010; Hoskins and Ambrizzi, 1993; Wettstein and Wallace, 2010; Wirth et al., 2018). In winter the North Atlantic jet stream can assume five different configurations (southern, central, northern, tilted, and mixed, Madonna et al., 2017) with distinct patterns of storm propagation, all of which are associated with regional climate anomalies over Europe.

The three classification methods for circulation regimes that we use in this study are not independent: the strength and position of the jet are intrinsically linked to changes in SLP and thus to the NAO and blocking; blocking over Greenland, Scandinavia and the Iberian Peninsula correspond to a southern, mixed, and northern configuration of the jet stream, respectively (Madonna et al., 2017). However, the NAO does not map clearly onto distinct jet configurations or blocking patterns, with the positive phase being especially ambiguous (Woollings et al., 2010; Davini et al., 2014; Madonna et al., 2017).

These classifications of North Atlantic atmospheric variability thus offer closely related but alternative views of seasonal climate fluctuations. For example, an extreme season may occur due to the unusual persistence or frequency of a certain jet configuration (Madonna et al., 2019), without a corresponding extreme value of the NAO index or blocking pattern. This study aims to compare the ability of three classification methods (NAO, blocking, jet configurations) to reconstruct seasonal climate anomalies over Europe. By knowing the frequency of each circulation pattern, we assess the skill of each method to reproduce the sign (i.e. correlation) and amplitude (i.e. ratio of standard deviations) of seasonal precipitation and temperature anomalies.

## 2 Methods and data

We focus on the low-level wind (900 to 700 hPa), two-metre temperature, and total precipitation. The analyses are conducted for winter (DJF; the 90 day period December 1 –February 28), with ERA-Interim reanalysis (Dee et al., 2011) for the period 1979-2014, interpolated to a 0.5° horizontal resolution. To identify blocking, 6 hourly data of geopotential height at 500 hPa are used. For the rest of the analysis, daily means are used.

### 2.1 Classifications

#### 2.1.1 NAO

We use the daily NAO time series from NOAA (downloaded from ftp://ftp.cpc.ncep.noaa.gov/cwlinks/norm.daily.nao.index. b500101.current.ascii), which is calculated using geopotential height at 500 hPa and covers the whole North Atlantic basin north of 20°N (green box in Figure 1). A day is classified as a positive (negative) NAO day if its NAO value is above 0.5 (below -0.5) of the wintertime NAO standard deviation. The remaining days, about 35 days per winter (Table 1), are considered neutral NAO days.

#### 2.1.2 Blocking

Blocking events are identified on 6-hourly data following the criteria from Scherrer et al. (2006), which define a block as a reversal in the meridional gradient of the geopotential height at 500 hPa in a 30° latitudinal band that lasts for at least 5 days. Climatologically in the North Atlantic there are three main regions affected by blocking (shading in Figure 1), one over Greenland, one over northern Europe/Scandinavia, and one offshore of the Iberian Peninsula. Note that the enhanced frequency at 30°N is an artefact of the detection method (see discussion in Davini et al., 2014). We define three boxes to capture these three regions: Greenland (65-25°W, 60-75°N, GB, orange box in Figure 1), Scandinavia (15°W-25°E, 50-65°N, SBL, red box) and offshore of the Iberian Peninsula (30°W-0°, 40-50°N, named as in Davini et al. (2014) Iberian wave breaking (IWB), blue box).

A day is considered a blocked day if at least 10% of the gridpoints in the respective box satisfy the blocking criteria. This choice reduces the number of blocking events identified as it detects blocking episodes at a later stage of their development (i.e. when they are spatially larger) than if no threshold was applied, and avoids the detection of blocks that are predominantly located upstream or downstream of the boxed regions. Table 1 reports the average (standard deviation) number of days per winter with blocking at different locations. During the 35 winters, we detect on average 13.5 days per winter of GB, 16.9 of SBL and 13.0 of IWB while 51.9 days are considered as "non blocked" (NB). It can occasionally happen that during one day several regions are simultaneously blocked, therefore the sum of blocking and NB days do not sum to exactly 90 days (i.e. one winter).

### 2.1.3 Jet clusters

Cluster analysis applied to SLP or geopotential height is commonly used to classify circulation patterns into so-called weather regimes. In the North Atlantic during winter, four classical regimes are identified (e.g Cassou, 2008; Michelangeli et al., 1995; Vautard, 1990), although the optimal number of clusters is not clear-cut, and depends somewhat on the algorithm, atmospheric field, domain, and considered temporal period (Christiansen, 2007; Dorrington and Strommen, 2020; Falkena et al., 2020; Woollings et al., 2010). Madonna et al. (2017) showed that applying a cluster analysis to the low-level jet leads to four jet configurations that correspond to the four classical weather regimes in the Euro-Atlantic sector: the northern jet resembles the Atlantic Ridge regime, the central jet resembles the Zonal/NAO+ regime, the mixed jet resembles the Scandinavian Blocking regime, and the southern jet resembles the Greenland Anticyclone/NAO- regime (see Figure 8 in Madonna et al., 2017). Using five clusters, the zonal regime can be further separated in a central and tilted jet (see Figure 10 in Madonna et al., 2017). In this study, we use five jet clusters, which gives more distinct jet configurations over the North Atlantic.

We calculate the daily mass-weighted average zonal wind (U) between 900-700 hPa in the sector 60°W-0°, 15-75°N (black box, Figure 1) and use this field to calculate jet clusters as described in Madonna et al. (2017). We perform an EOF analysis on the low-level wind to reduce the dimensions and apply a k-mean clustering algorithm to the first five principal components, which explains up to 80% of the wintertime variability in that sector.

Every day is associated with a cluster depending on the normalized inverse Euclidean distance ($d$) from the cluster centroid (i.e. $d = 1$ at the centroid location and $d = 0$ far from the centroid) in the five-dimensional space of the principal components. The Euclidean distance ($E$) from the centroid $c$ for a point $x$ in a n-dimensional space is defined as $E(c, x) = \sqrt{\sum_{i=1}^{n}(c_i - x_i)^2}$. $E$ is normalized such that the proximity measure $d = 1/E$ sums to 1 over all days. Since some days can be close to more than one centroid, in particular during transition days from one cluster to the other, we keep only days whose $d$ to the respective centroid is larger than $0.5$. A sensitivity analysis suggests that the choice of the threshold $d$ does not have a significant impact on the results (see supplement). Approximately 54% of the 3150 (=35 x 90) days are unequivocally attributed to a specific cluster (see Table 1, 46% of days are not assigned to any cluster).

The five clusters represent a southern jet (S-jet), a central jet (C-jet), a northern jet (N-jet), a tilted jet (T-jet), and a mixed jet (M-jet with a split structure). On average each cluster occurs between 7.3 (S-jet) and 12.0 (T-jet) days per winter, as reported in Table 1, with large winter-to-winter variability (standard deviation).

### 2.1.4 Associated weather anomalies

Each NAO phase, blocking category and jet cluster is characterized by different circulation, precipitation, and temperature anomalies. We compute daily anomalies of zonal wind, precipitation, and temperature and then construct composites for each NAO phase, blocking category, and jet cluster by averaging the daily fields of all (defined) days belonging to the corresponding category. For wind and precipitation the daily anomalies are calculated by subtracting the climatological 35-year winter mean, as previous studies have shown a weak seasonal cycle of those variables within DJF (Woollings et al., 2014; Zveryaev, 2004).

However, the seasonal cycle is much stronger for temperature and therefore temperature anomalies are calculated by removing the daily 35-year average, smoothed with a 30-day running mean. The resulting anomalies are presented in section 3.1.

## 2.2 Seasonal Reconstructions

To reconstruct the seasonal anomalies we count the number of days in each NAO phase, blocking category and jet cluster for each winter. Similar to Cortesi et al. (2019), we then reconstruct the seasonal precipitation and temperature anomaly maps $A_{rec}(\phi, \lambda, t)$ for each season ($t$) as follows:

$$A_{rec}(\phi, \lambda, t) \quad = \quad \sum_i (Y_i(\phi, \lambda) \cdot f_i(t)) \tag{1}$$

where $\phi$, $\lambda$ and $t$ are latitude, longitude and time, respectively; $i$ represents the two NAO phases, three blocking categories or five jet clusters; $Y_i(\phi, \lambda)$ are the maps of seasonal average precipitation/temperature anomalies associated with the considered pattern; and $f_i(t)$ is the fraction of time the pattern occurs in the given season $t$ (i.e. $f_i(t) = \frac{\#\text{days of pattern } i}{\#\text{days per season (=90)}}$).

For example for the jet clusters in DJF 2013/2014 (cf. Figure 5) there are 0 days classified as S-jet and M-jet, 11 days as T-jet, 7 days as N-jet and 60 days as C-jet. The remaining 12 days belong to the undefined category and are not used. The reconstructed anomaly using the jet clusters for DJF 2013/2014 is:

$$A_{rec}(\phi, \lambda, 2013/2014) \quad = \quad \frac{0 \cdot Y_S(\phi, \lambda) + 0 \cdot Y_M(\phi, \lambda) + 11 \cdot Y_T(\phi, \lambda) + 7 \cdot Y_N(\phi, \lambda) + 60 \cdot Y_C(\phi, \lambda)}{90} \tag{2}$$

where the $Y_i(\phi, \lambda)$ are the composite maps of precipitation or temperature anomalies and the subscript stands for the jet type.

This reconstruction method assumes that the average of the undefined days represents the climatological mean; i.e. the average anomaly field associated with undefined days is close to zero. In the supplementary material (Figure S1) we show that this is indeed the case. In the case of the residual not summing to zero, equation (1) must be modified to include the pattern and the fraction of time of the unclassified days. To compare the ability of each classification method to reconstruct seasonal anomalies we compute at each grid point the correlation as well as the coefficient of efficiency (CE, described in the next section) between the reconstructed and the observed (ERA-Interim) seasonal anomalies.

### 2.2.1 Coefficient of Efficiency (CE)

Assume $o$ is the observed quantity and $p$ is the reconstructed quantity. The Coefficient of Efficiency (CE) (Nash and Sutcliffe (1970), see also Bürger (2007); Briffa et al. (1992); Wang et al. (2014)), which we calculate at each gridpoint $(\phi, \lambda)$, is given by

$$CE \quad \equiv \quad 1 - \frac{\sum\limits_t (o_t - p_t)^2}{\sum\limits_t (o_t - \overline{o})^2} \quad , \tag{3}$$

where $\overline{(\ )}$ denotes the mean of a quantity and the sum is over all winter seasons ($t$).

Equation 3 can be rewritten as:

$$CE \quad \equiv \quad 1 - \frac{\sigma^2(o') + \sigma^2(p') - 2<o',p'>/N + (\overline{o} - \overline{p})^2}{\sigma^2(o') + \overline{o}^2} \quad , \tag{4}$$

where $(\ ')$ denotes the anomaly about the mean $\overline{(\ )}$ and $\sigma$ is the standard deviation.

If we consider only anomalies and therefore assume that the mean of $o$ and $p$ are both zero, the equation can be simplified
such that

$$CE \quad \equiv \quad 2ra - a^2 \quad , \tag{5}$$

where $a$ is an amplitude ratio of the standard deviations of the time series,

$$a \quad = \quad \frac{\sigma(p)}{\sigma(o)} \tag{6}$$

and $r$ is the correlation between $o$ and $p$.

In our case, $o$ is the observed anomaly of precipitation/temperature and $\overline{o}$ is by definition zero, while $p$ is the reconstructed anomaly and $\overline{p}$ is not necessarily zero (see Figure S2). In this study, the CE is calculated using equation (3), while the simplification (eq. 5) is used for scaling the anomaly amplitudes (see Sec. 3.3).

When applied to reconstructions and observations, the CE is a measure of skill in reconstruction that is more restrictive than a simple correlation because it penalizes for both phase and amplitude misfits. For a perfect reconstruction, CE=1. For a
reconstruction with observed variance (a=1) that is correlated with the observed time series of seasonal anomalies at r=0.5, the result is CE=0. For a reconstruction that is perfectly correlated with observations but with twice the observed amplitude, we arrive at the same result of CE=0. We consider CE>0.25 to indicate a good reconstruction. A reconstruction that has perfect variance (a=1) and CE= 0.25 would explain 39% of the observed variance (r=0.63); given 35 years of data (degrees of freedom), this correlation would be significant at p=0.0001. For a reconstruction with perfect correlation (i.e. r=1), CE>0.25
can be obtained for amplitude values 0.13< $a$ <1.87. As the CE maximizes when $r = a$ (and max(CE)= $r^2$), a CE of 0.25 also implies a minimum correlation of 0.5 independent of the amplitude error. For this reason, only points with correlation greater than 0.5 are considered when optimizing the CE (see Sec. 3.3).

## 3    Results

### 3.1    Classification of anomalies and interannual variability

Distinct wind, temperature and precipitation anomalies are associated with each NAO phase, blocking category and jet cluster (Figs. 2-4). Figure 2 shows the zonal wind anomalies for each pattern (colours) and the climatological DJF zonal wind distribution (black contours). Figures 3 and 4 show precipitation and temperature anomalies (colours) respectively, and the composite

zonal wind (black contours). The panels are arranged such that each row includes "similar"[1] patterns identified using the NAO index (left column), blocking (centre column) and the jet clusters (right column). In general, wind and temperature patterns along each row are remarkably similar given the composites including different numbers of days; e.g. for the top row, there are on average 25.4 days per season corresponding to the NAO- group, 13.5 days to GB, and 7.3 days to S-jet (see Table 1). Despite similarity in the spatial structures of the composites, there can be large differences in the strength of the anomalies for the different classification methods.

The top rows of Figures 2-4 show a clear correspondence between the negative NAO phase, Greenland blocking (GB), and the S-jet cluster, consistent with previous work (Woollings et al., 2010; Madonna et al., 2017). In all three composites the jet is located southwards of its climatological position (Figure 2). This southerly shifted jet is also zonally oriented, which can be seen in the total composite wind fields shown in black contours in Figures 3 and 4. In all three composites, precipitation is enhanced in the jet core and at its ends, e.g. over the Iberian Peninsula (Figure 3, blue shading). Although the patterns look fairly similar, they differ in intensity, with the highest precipitation anomalies found in the S-jet cluster. When the jet is shifted south (NAO-, GB, S-jet), Greenland is warmer than usual, while northern Europe and the Barents Sea are colder than usual (Figure 4, shading). The anomaly patterns shown in the top row resemble the Greenland Anticyclone/NAO- regime in the framework of the four classical Euro-Atlantic weather regimes (e.g. cf. with Cattiaux et al., 2013; van der Wiel et al., 2019).

There is less correspondence between the positive NAO phase and any other pattern. As the positive NAO phase has often been referred to as the unblocked or unperturbed state (e.g. Woollings et al., 2008, 2010) it does not resemble any of the blocking patterns. The positive NAO composite has wind and temperature anomaly patterns reminiscent of both the tilted and central jet clusters (Figs. 2 and 4, shadings), with a jet shifted to the north (and tilted) and warmer temperatures over central and northern Europe. The warm anomaly over the Barents Sea observed during the positive NAO phase is weakly present in both the tilted and central jet composites, while the cold anomaly over Greenland is linked to the tilted jet rather than the central jet. Also, precipitation anomalies of the positive NAO phase are more similar to the tilted jet cluster than the central jet cluster, indicative of the large change in precipitation pattern associated with the relatively small shifts in jet position.

Blocking over the Iberian Peninsula (IWB) is associated with a northward shift of the jet (N-jet), while blocking over Scandinavia (SBL) splits the jet, resulting in a M-jet configuration Madonna et al. (2017). Translated into the four classical weather regimes, blocking in these regions thus occurs during the Atlantic Ridge and Scandinavian blocking regimes, respectively (Madonna et al., 2017). The blocked region is drier than climatology for both cases (Figure 3), with less precipitation offshore and over the Iberian Peninsula associated with the N-jet cluster (IWB), and less precipitation over central Europe associated with the M-jet cluster (SBL). During IWB, southern Europe and northern Africa are colder, while northern Europe is warmer than normal; during N-jet, only the cold anomaly is evident. During SBL, most of Europe is cold and northern Scandinavia is warm.

The zonal wind anomaly composites of the days that are not included in any category are close to zero for the NAO (neutral days) and for the jet (undefined), but not for the non-blocked (NB) category (see Figure S1). This behaviour can be understood from the point of view of the classical weather regimes, as the non-blocked category includes days that belong to

---

[1]based on temporal correlation shown in Figure 5

the "zonal/NAO+" regime as well as days with weak wind anomalies. Therefore, in terms of zonal wind, the composite of non-blocked days is very different from climatology. However, this effect is less evident in the precipitation and temperature anomalies (i.e. anomalies close to zero).

The correspondence between the jet and blocking composites extends to seasonal time scales. Figure 5 shows the time series of the occurrence (i.e. the number of days per winter) of each NAO phase, blocking and jet cluster. The time series of the S-jet cluster has a correlation of 0.64 with GB and 0.67 with the negative NAO phase. However, the S-jet is less frequent than the other two, with on average only 7.3 days per winter, compared to 13.5 days of GB and 25.4 days of the negative NAO phase (Table 1). The N-jet occurs on average 8.6 days per winter and IWB 13.0 days (Table 1) and their time series (Figure 5b) are

correlated at 0.71. The M-jet occurs on average 9.7 days, SBL 16.9 days and their time series are correlated at 0.58. The C-jet (10.9 days) and T-jet (12.0 days) are the most frequent jet clusters, however, they show a relatively low correlation with the positive NAO time series (0.30 and 0.45, respectively).

The winter to winter differences in the number of days in each jet cluster and blocking type is large: standard deviations are of similar magnitude to the mean values (Table 1). Moreover, about 40-60% of the days are classified as NAO neutral days,

unblocked or undefined (for the jet cluster). The composites of precipitation and temperature for those categories are similar to climatology and therefore their patterns are characterized by little anomalies, in particular over the European continent (Figure S1).

### 3.2 Seasonal reconstructions

Using the method described in section 2.2, we reconstruct seasonal anomalies of temperature and precipitation from each of the

220 three classification methods: the NAO, blocking, and jet clusters. Based on our definitions, for each classification the average number of days per season used for reconstruction is between 38.1 (blocking composites) and 55.4 (NAO composites) days (Table 1). We compare our reconstructed seasonal anomalies to the observed anomalies to evaluate the skill of each method for precipitation and temperature.

### 3.2.1 Correlation

The ability of each method to reconstruct seasonal weather anomalies varies greatly with location. To assess this, we calculate for each gridpoint the correlation coefficient between the time series of seasonal reconstructed anomalies and that of the actual anomalies from the ERA-interim reanalysis. Figure 6 shows the spatial distribution of this correlation coefficient over Europe for precipitation (left column) and temperature (right column); regions are masked with white dots when the correlation coefficient is below 0.5. The spatial structure of correlations for temperature is much smoother than that for precipitation; this

is consistent with smoother variations in temperature fields relative to precipitation, which varies at much smaller spatial scales. For the reconstructions based on jet clusters, precipitation agrees better (higher correlations) with observations than temperature, in particular in regions of high topography. In these regions it is likely that precipitation depends more uniquely on circulation than does temperature, as temperature can be affected by other mechanisms including cloud cover and land

surface feedbacks. The correlation coefficients for wind are much higher than those of precipitation and temperature for all classifications (Figure S3).

Overall, the correlations between precipitation reconstructions and observations are higher in western Europe and Scandinavia, and lower in central to south-east Europe. Regions with low correlations also show little seasonal variability (i.e. small seasonal standard deviations, Figure S4) suggesting that large-scale circulation patterns have less impact on precipitation in these regions. There are relatively minor differences in the correlation of precipitation for the different methods, although correlations over France are noticeably worse in the NAO reconstruction (cf. figures 6a with 6c and e).

The skill of the temperature reconstructions depends greatly on the classification method. Over Spain and France, the blocking method does substantially better than the NAO and slightly better than the jet clusters. Conversely, the NAO performs much better in a band from 50 to 65°N than the other methods, but substantially worse south of 50°N. This is consistent with the temperature anomalies in Figure 4 - neither positive nor negative NAO is associated with strong temperature anomalies across southern Europe. Winter temperature variability exhibits a southwest-northeast gradient (Figure S4) and regions with larger variability (e.g. Scandinavia) often exhibits larger correlations for all classification methods.

### 3.2.2 Coefficient of Efficiency

Having shown strong correlations between reconstructed and observed seasonal anomalies for many regions of Europe, we now examine the coefficient of efficiency (CE), which takes into account both the correlation (i.e. the phase) and the magnitude of the reconstructed values relative to observations. The spatial pattern of the CE for precipitation (Figure 7) generally follows that of the correlation (Figure 6), but the absolute values are lower – less than 0.25 across much of the domain. The highest CE values are for the jet classification, in particular over Iberia, France and Norway, while all methods have low skill over central Europe. Interesting is the poor CE performance for blocking over most of the domain, with even negative CE values in regions where the correlation is above 0.5 (non-dotted regions). Considering the CE definition expressed by equation 4, we see that a non-zero mean of the reconstructed anomalies ($\bar{p}$) can influence the CE values (note that the mean of the observed anomalies ($\bar{o}$) is by definition zero). The mean reconstructed precipitation ($\bar{p}$) is approximately zero for the NAO and the jet clusters, but not for blocking (Figure S2), partly explaining the lower performance of the latter classification. Another reason for low CE could be the underestimation of the amplitude of the reconstructed anomalies, as we will discuss later (see section 3.3).

The CE is typically substantially lower for temperature than for precipitation (Figure 7). The NAO classification does better in northern Europe, while the other two classifications have more skill in southern Europe, in particular over Iberia. For purposes of applicability, we focus on land regions over Europe. In the supplementary material (Figure S5) we show plots that extend westward into the North Atlantic; values of CE are typically higher over the ocean off the west coast of Europe and the differences in skill for the three methods become even more apparent, e.g. over the North Atlantic the NAO can not reconstruct precipitation anomalies in the 45-50°N latitudinal band, while the blocking has the best temperature reconstruction over North Africa. The CE values for zonal wind (shown in Figure S5) are much higher than those of precipitation and temperature, which is not surprising as all classification methods are based on circulation anomalies. For zonal wind, high CE skill is concentrated in two latitudinal bands for the NAO and blocking, while the jet classification has skill over the whole North Atlantic.

### 3.3 Scaling factor beta

The reconstruction described in section 2.2 assumes that the composite mean precipitation or temperature field for each category is representative of all the days falling into the composites. This assumption works well for variables that follow a Gaussian distribution. However, the assumption does not necessarily hold for fields such as precipitation, which is known to be skewed. Alternative approaches include using the median instead of the mean to define the anomaly patterns, which lessens the influence of extreme values, or estimating the representative anomaly values from a random sample within each category as done by Fereday et al. (2018).

We opt for a different method whereby we adjust the reconstruction *a posteriori* based on estimates of the anomaly values that best represent each pattern. We start with the approximation of the CE expressed as a function of correlation $r$ between the reconstructed and observed time series at each gridpoint, and amplitude ratio $a$ of their standard deviations (Eq. 5). It is not possible to improve the correlation $r$ but it is possible to adjust the amplitude ratio $a$ to boost the CE. We do so by calculating the "centre of mass" of the CE in correlation-amplitude space (with each gridpoint having a weight of 1), then determining the scaling factor $\beta$ that moves the centre of mass in the y-direction so that $a = r$, which maximizes the CE. Thus, the seasonal anomaly of precipitation or temperature from equation (1) is:

$$A_{rec}(\phi, \lambda, t) \quad = \quad \beta \quad \sum_i (Y_i(\phi, \lambda) \cdot f_i(t)) \tag{7}$$

For example, for the NAO, the unscaled precipitation reconstruction (Figure 7a) is shown in $a-r$ space by the black contours in Figure 8a, with almost all amplitude ratios falling below the $a = r$ line (red). In other words, the amplitudes of the reconstructed anomalies tend to be underestimated, even when there is relatively good correlation (right edge of area outlined by black contours). Applying the scaling factor increases the amplitude of these anomalies such that the scaled reconstruction (shown in $a-r$ space by the blue filled contours) has a centre of mass that lies on the $a = r$ line at higher values of CE (white contours).

Only points with a correlation above 0.5 (which represents approximately the 1% significance level of a two-tailed t-test with 34 degrees of freedom) are used to calculate the scaling factor; this prevents amplitude biases from points with weak correlations, and thus little skill, from affecting the reconstruction ability of points with higher skill. The approximation of the CE given by equation (5) requires that the mean of the reconstructed anomalies ($\overline{p}$) be close to zero. This assumption is valid for the NAO and jet clusters, but not for the blocking (see Figure S2). Therefore, the scaling factor $\beta$ is calculated only for the first two classifications. The need for a scaling factor to maximise the reconstruction skill is discussed further in section 4.

Table 2 gives the scaling factors for the reconstructions based on the NAO and jet clusters, indicating that the amplitudes for all variables (precipitation, temperature, and wind) are underestimated by approximately 50% ($\beta \sim 2$). Comparing the CE skill score for the unscaled ($\beta = 1$) reconstructions (Figure 7) versus the scaled reconstructions (Figure 9), we see that the scaling substantially improves the seasonal temperature reconstructions over most of the domain, while the improvements are more localized for precipitation.

The skill of the seasonal reconstructions is perhaps most easily illustrated for specific locations of interest across Europe (Galicia in Spain, Berlin, Bergen and London, indicated by X1-4 in Figure 6a-b). Figure 10 shows the observed precipitation and temperature anomalies (black, ERA-Interim) and reconstructions for these locations (scaled for the NAO in grey and jet clusters in red, unscaled for blocking in blue; the unscaled anomaly magnitudes are expected to be underestimated). Precipitation is well reconstructed for Galicia (10a), Bergen (10e) and London (10g) using the five jet clusters (red): correlations between reconstructed and observed anomalies range from 0.76 to 0.87. None of the methods reconstruct well the seasonal precipitation anomalies in Berlin (10c); this is not surprising, given there is little variability in the seasonal averaged precipitation. In terms of temperature, the three methods produce skilfull temperature reconstructions for all locations but Galicia, where the NAO has no skill.

## 4 Discussion

We have shown in this paper that the skill of various classification methods in reconstructing European seasonal surface precipitation and temperature anomalies is strongly dependent on the region. There is no one method that works best for all regions and variables, and to maximise the coefficient of efficiency of the seasonal reconstructions a scaling factor of approximately 2 is required.

Considering correlation (Figure6) and the unscaled CE (Eq. 3 and Figure 7), one might expect the skill of a reconstruction to improve with the number of basis functions (patterns) used. For example, more of the interannual variability should be captured by the jet clusters (five patterns) than blocking (three patterns) or the NAO (only two patterns). While this is mostly true for precipitation, it does not always apply for temperature. In northeast Europe and Scandinavia, the NAO outperforms the other classification methods. A possible explanation might lie in the different domains used to define the classification patterns: the region used for the jet clusters is much smaller than that for the NAO (Figure 1), with an eastern limit at the Greenwich meridian (0 °) for the jet cluster while the domain used to define the NAO extends 30° further east and includes Europe. Thus, it is not so surprising that the NAO is better able to capture the seasonal anomalies over central and eastern Europe, as circulation variability over these regions is integrated into the NAO definition. In fact, it is rather remarkable that so much of the seasonal precipitation and temperature signal over Europe can be inferred just by knowing the circulation over the North Atlantic ocean (i.e. using the jet clusters). In some regions the seasonal anomalies reconstructed from the blocking patterns are similarly or even more skillful in term of correlation than those using the jet regimes (e.g. precipitation over France), but they have much lower CE scores because the sum of the residual anomaly pattern is not zero (see Figure S2). This behaviour can be partly understood knowing that non-blocked days encompass days with weak winds as well as days with strong zonal wind; the composite mean of those days (Figure S1) might sum up to zero over 35 years, but this does not have to be true for a season.

One might also expect the skill of a reconstruction to depend on the amount of information included, i.e the number of days per season used. In our reconstructions, we include only the days that distinctly belong to a certain basis function within each classification method. Interestingly, this sums to about half of the total days in the record, regardless of method. One could

of course use more information, but this does not necessarily improve the reconstruction skill. Indeed, a sensitivity test (see Supplement Section A) shows that including all days for the jet classification leads to very little improvement in the CE score. A similar example is shown by Fereday et al. (2018), who used 30 SLP patterns (i.e., many more basis functions) and all days to reconstruct winter precipitation. The correlation between their reconstructed and the observed precipitation was of $\sim$0.8 over northern and southern Europe. Averaging our results over the same regions, we obtain lower but comparable correlations between observed and reconstructed precipitation (0.54-0.78 for the northern region (Figure S6, blue) and 0.53-0.66 for the southern region (Figure S6, red)).

All the reconstructions underestimate the amplitude of the observed precipitation and temperature anomalies. This contributes to the low CE skill score in many regions in Europe, despite relatively high correlations between the reconstruction and reanalysis data. The CE skill score for precipitation and temperature reconstructions for the NAO and jet improves when we introduce a scaling factor ($\beta$) of $\sim$2 to the composite mean anomaly patterns. A possible reason for the underestimation of the reconstructed amplitude is the large variability within each composite. For example for an S-jet day, we expect precipitation to be enhanced over the Iberian Peninsula (cf. Figure 3), but the exact location where the precipitation peaks, which is likely linked to the passage of a specific cyclone, varies from case to case (i.e. cyclones do not have the exact same path). Therefore, at each grid point the standard deviation within each composite can be quite large (Figs. S7 and S8). To gain some insight into why this scaling factor is required, we repeat the NAO seasonal temperature reconstruction using regression techniques instead of composites. Using a simple ordinary linear regression on daily NAO and temperature values we find relationships (°C per unit anomalous NAO index) very similar to those found by the composite method shown in Figure 4. A reconstruction is then made by multiplying the regression pattern by the mean NAO value for each season. In this regression approach, all days are used in the reconstruction (compared to about half the days in the composite approach). However, the correlation and CE values of the two reconstructions are very similar, and both require a scaling factor to maximise the skill. This suggests that the need for a scaling factor is not linked to the omission of information (i.e. the neutral NAO or undefined days).

If, instead, the regression between daily temperature anomalies and daily NAO values is calculated using a weighted orthogonal distance regression (using the python package scipy.odr), the regression relationship changes - the slope of the linear fit generally increases. An example is shown in Figure 11 for Bergen (cf., blue regression line for ordinary least squares, black regression line for orthogonal distance). This increase in regression slope means that the reconstruction amplitudes increase, and there is less need for a scaling factor. The weighted orthogonal distance regression takes into account "noise" in both the temperature and the NAO values, while ordinary least squares considers the values of the independent variable (in this case the NAO) to be exact (e.g. Wu and Yu, 2018). This noise may be related to a lag/lead relationship between circulation patterns and surface weather anomalies and/or uncertainty in the connections between the NAO circulation anomalies and surface temperature. In Figure 11 the mean composite values for this grid-box are shown by the cyan markers, and they fall on the ordinary least squares regression line. The median composite values are also shown in purple, demonstrating that using the composite median would not remove the need for the scaling factor. This suggests that the need for the scaling factor in the composite and ordinary least squares regression reflects the large variability in temperature and precipitation within each classification pattern (cf. Figure S7 and S8).

Finally, the ability to reconstruct precipitation and temperature might be affected by extreme events. For example, if a single extreme precipitation event is responsible for the lion's share of precipitation in a specific season, we would not expect a skilful reconstruction of the seasonal anomaly using the average precipitation signals associated with each basis function. The effect of extreme events varies regionally, as shown for summer temperatures by Röthlisberger et al. (2020). It would therefore be interesting to investigate to what extent extreme events influence the seasonal precipitation and temperature anomalies over Europe and how these events are related to circulation anomalies.

## 5 Concluding remarks

In this study, we investigate how well seasonal anomalies in European precipitation and temperature can be reconstructed based on the frequency of circulation patterns defined using three different classifications: the NAO index, blocking, and the configuration of the North Atlantic jet stream.

The skill of the various classifications in reconstructing seasonal anomalies depends on the variable and region of interest. For the NAO and jet clusters, the regions of high skill for precipitation are rather different than for temperature (see Figure 9). Precipitation in western Europe is particularly well reconstructed, with many coastal and mountainous areas showing coefficient of efficiency values for scaled precipitation greater than 0.5 (Figure 9). For these areas, precipitation in winter is directly linked to the propagation of storms travelling from the Atlantic (Hawcroft et al., 2012; Pfahl et al., 2014), which is to first-order set by large-scale circulation variability. Still, the relationship between circulation and precipitation is far from straightforward, seen by the fact that in some places precipitation is reconstructed with comparable (high) skill by all three methods (e.g., Bergen, Norway), while in other places one method performs worse than others (e.g., NAO in Galicia). For temperature, circulation influences the horizontal and vertical advection of air, allowing a simple index like the NAO to provide skilful reconstructions across much of northern Europe. However, over France temperatures are better captured by the blocking reconstruction. In southern and inland regions (e.g., Berlin, Germany), none of the methods provides skilful reconstructions of temperature or precipitation, suggesting that factors unrelated to circulation are important, for example radiative forcing (e.g. clear vs. cloudy, Trigo et al., 2004), soil moisture coupling (Fischer et al., 2007) or snow-albedo feedback.

In the end, no single classification metric emerges as providing "the best" reconstruction of both precipitation and temperature across all regions. The three circulation metrics - jet clusters, blocking and NAO - are clearly connected, but emphasize different aspects of the large-scale flow with different implications for surface climate. The results presented here can provide guidance on which classification method is most suitable for linking regional climate to circulation variability. Through this approach, one may gain insight into the surface impacts of weather events over a range of time scales. Regime-based reconstruction may prove useful in extended range predictability (Kim et al., 2016; Scaife et al., 2014) and in assessing changes in the frequency of weather patterns that constitute the changes in the climatology under anthropogenic forcing.

*Code and data availability.* ERA-Interim data can be downloaded from the ECMWF page https://apps.ecmwf.int/datasets/data/interim-full-daily/ levtype=sfc/. The NAO index was downloaded from NOAA ftp://ftp.cpc.ncep.noaa.gov/cwlinks/norm.daily.nao.index.b500101.current.ascii. The method to identify blocking is described in Scherrer et al. (2006) and for jet clusters in Madonna et al. (2017). The winter time series

used in this study are available at DOI: 10.5281/zenodo.4011886.

*Author contributions.* DSB and EM designed the study. EM performed most of the analysis, with RHW contributing the NAO regression analysis. All authors contributed to the interpretation and discussion of the results and the writing of the paper.

*Competing interests.* CL and DSB are members of the editorial board of the journal.

*Acknowledgements.* EM and CL acknowledge funding from the Research Council of Norway (Nansen Legacy grant #276730). RHW re-

ceived funding from the European Union's Horizon 2020 research and innovation programme under the Marie Skłodowska-Curie Grant Agreement 797961, and from the Tamaki Foundation. ECMWF and NOAA are acknowledged for providing the ERA-Interim reanalyses and NAO data, respectively. We thanks two anonymous reviewers for their constructive comments that improved the quality of this manuscript.

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

**Table 1.** Average and (in brackets) standard deviation of the number of days per winter in each category. The positive (negative) NAO phase is defined as days with NAO values above 0.5 (below -0.5) of the time series standard deviation (see Section 2 for more information), and the remaining days are considered as neutral. We differentiate between blocking over Greenland (GB), over the Iberian Peninsula (IWB), Scandinavia (SBL) and non-blocked days (NB). The categories for the jet are: south (S-jet), tilt (T-jet), north (N-jet), mixed (M-jet) and central (C-jet). We also reported the number of days in the undefined category using the jet classification (undef).

| NAO | | blocking | | jet | |
|---|---|---|---|---|---|
| NAO- | 25.4 (16.8) | GB | 13.5 (11.4) | S-jet | 7.3 (7.7) |
| NAO+ | 30.0 (13.9) | | | T-jet | 12.0 (9.8) |
| | | IWB | 13.0 (12.3) | N-jet | 8.6 (7.5) |
| | | SBL | 16.9 (9.3) | M-jet | 9.7 (7.4) |
| | | | | C-jet | 10.9 (11.0) |
| neutral | 34.6 (11.1) | NB | 51.9 (14.0) | undef | 41.4 (10.0) |

**Table 2.** Scaling factors $\beta$ for two NAO phases, and five jet clusters for gridpoints over Europe (only land) and only with correlation larger than 0.5.

| scaling factor $\beta$ | precipitation | temperatures | zonal wind |
|---|---|---|---|
| two NAO phases | 1.8 | 1.9 | 1.9 |
| five jet clusters | 1.8 | 2.2 | 1.7 |

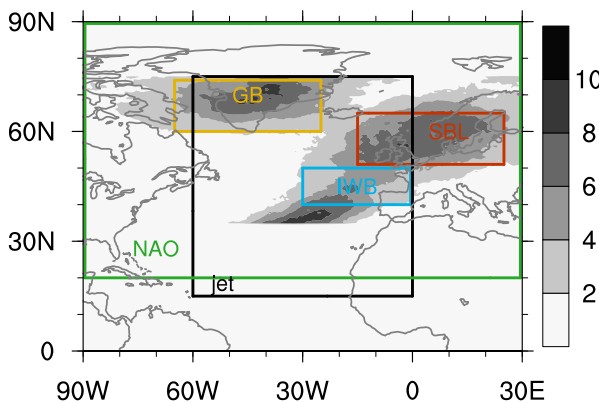

**Figure 1.** Winter (DJF) blocking climatology (shading, as % of time). The green and black boxes show the domain used for the definition of the NAO and jet clusters, respectively. The orange, blue, and red boxes denote the regions used for Greenland blocking (GB), Iberian wave breaking (IWB), and Scandinavian Blocking (SBL), respectively.

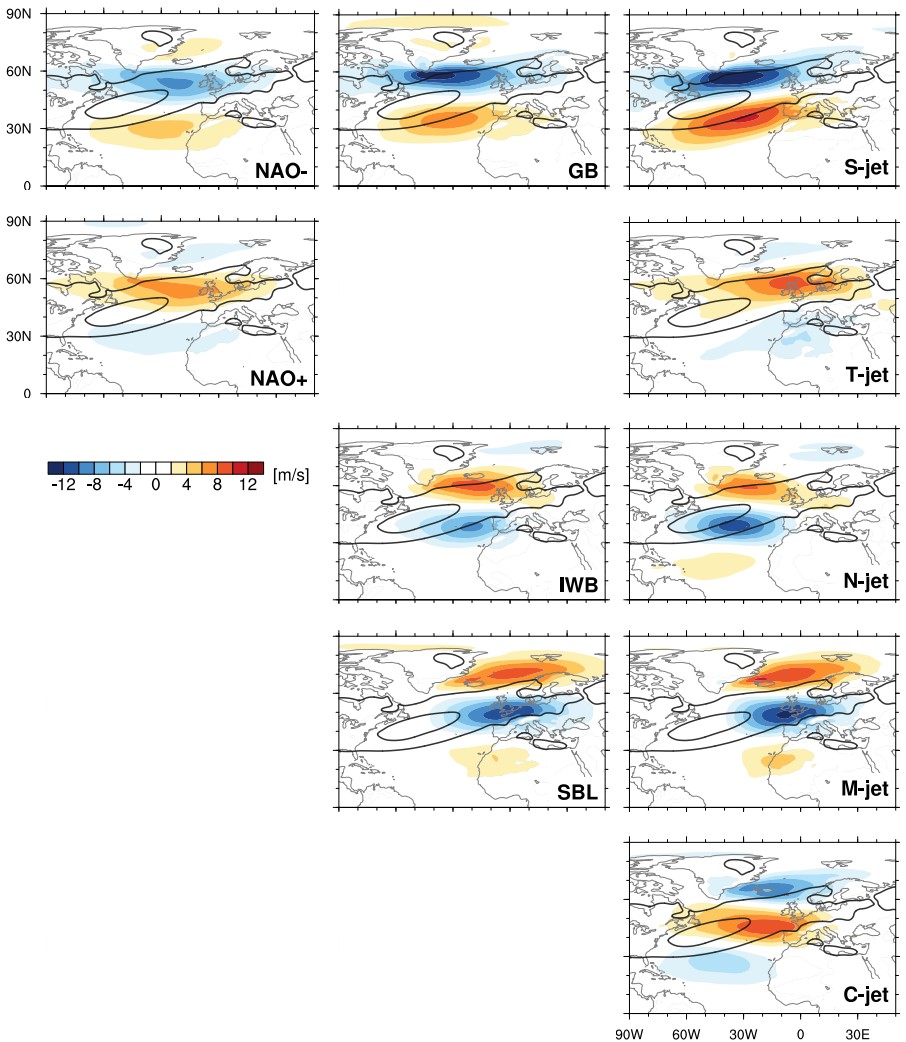

**Figure 2.** Zonal wind anomalies (shading, in m s$^{-1}$) at 850 hPa for two NAO phases (1st column), three blocking categories (2nd column) and five jet clusters (3rd column). Black contours show the climatological zonal wind at 850 hPa (contours at 5 and 10 m s$^{-1}$). The figure is organised such that maps in the same row represent similar circulation patterns identified by more than one method (NAO, blocking, jet regime).

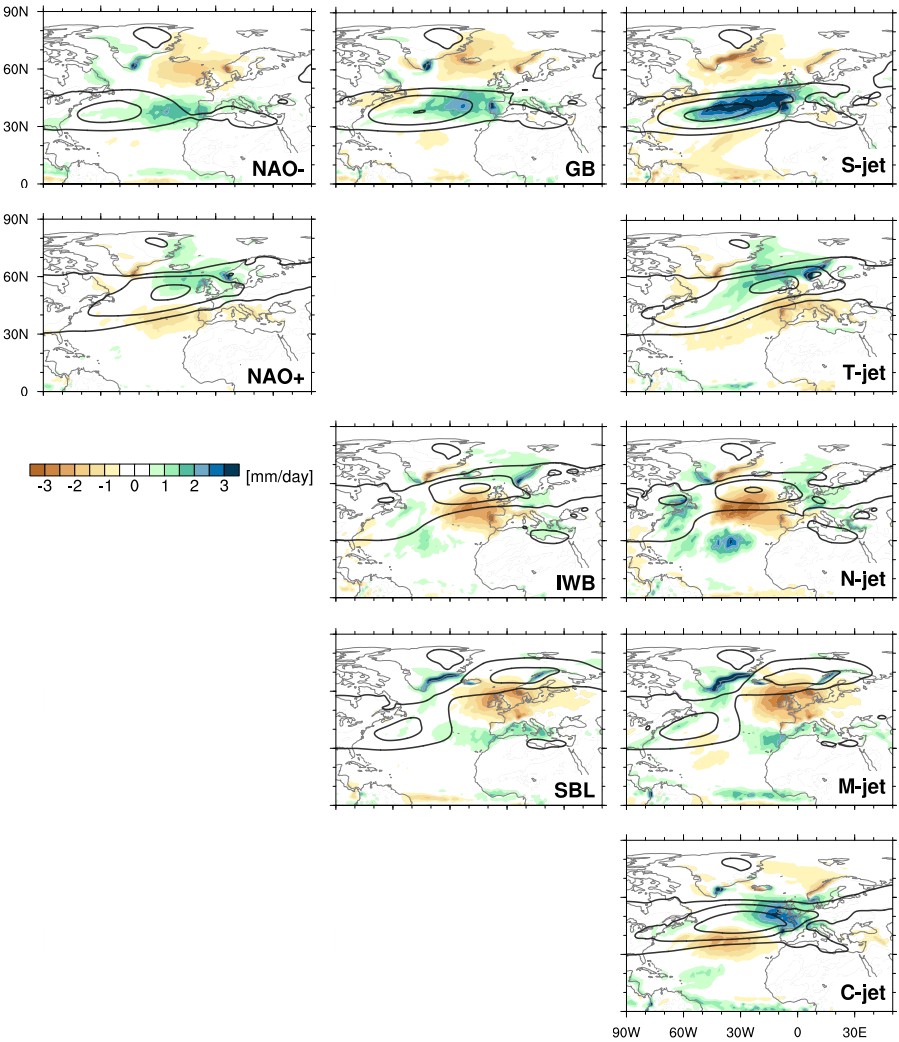

**Figure 3.** Precipitation anomalies (shading, in mm day$^{-1}$) and zonal wind at 850 hPa (contours at 5, 10, and 15 m s$^{-1}$) for two NAO phases (1st column), three blocking categories (2nd column) and five jet clusters (3rd column).

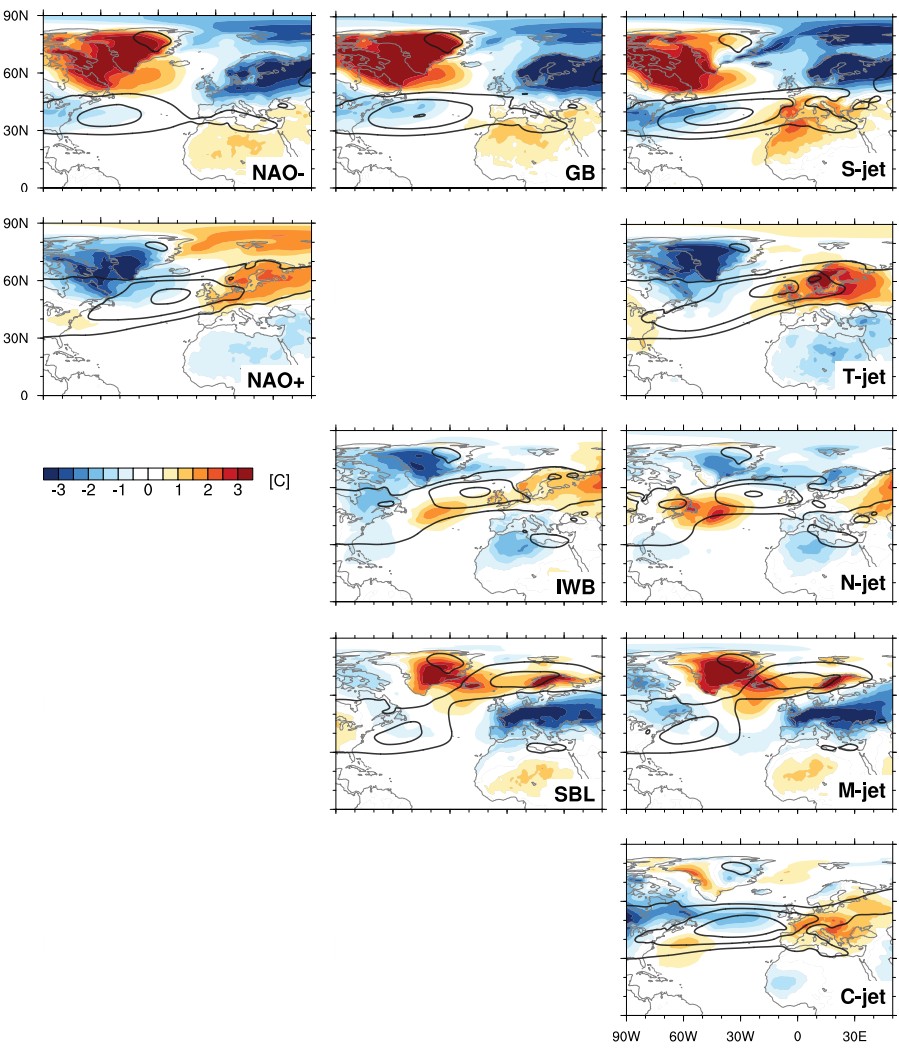

**Figure 4.** Two metre temperature anomalies (shading, in °C) and zonal wind at 850 hPa (black contours at 5, 10, and 15 m s$^{-1}$) for two NAO phases (1st column), three blocking categories (2nd column) and five jet clusters (3rd column).

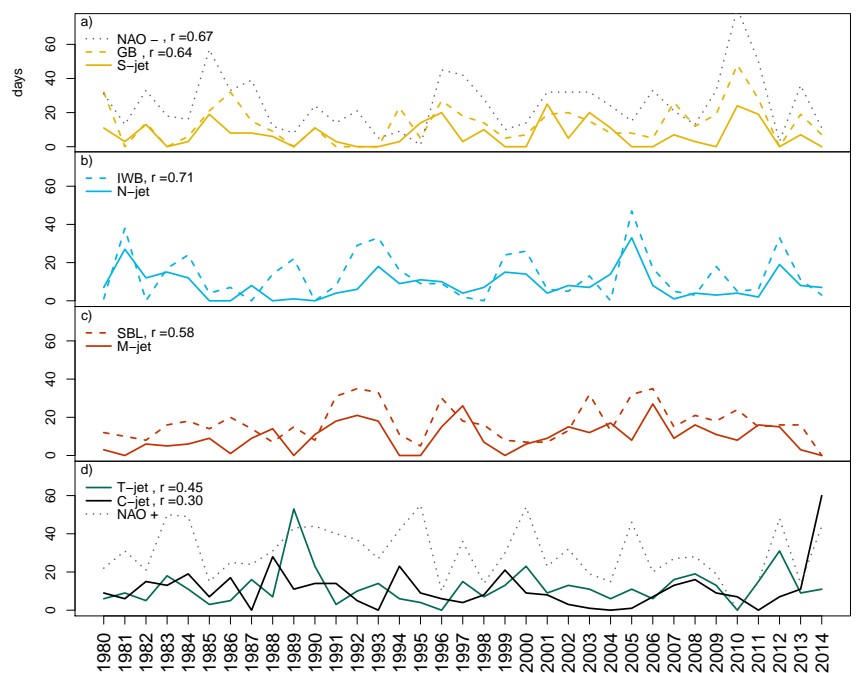

**Figure 5.** Time series of the number of days per winter for different categories. (a) negative NAO phase (NAO-), Greenland blocking (GB) and S-jet, (b) blocking over Iberia (IWB) and N-jet, (c) blocking over Scandinavia (SBL) and M-jet (d) positive NAO phase (NAO+), T-jet and C-jet. For a-c the correlation ($r$) between each time series of blocking and jet is shown in the plot. The negative NAO phase has a correlation of 0.67 with S-jet, while the positive NAO phase has correlations of 0.45 with T-jet, 0.30 with C-jet and 0.31 with N-jet. The year denotes the December-February period; e.g. 2013 is the average for December 2012 to February 2013.

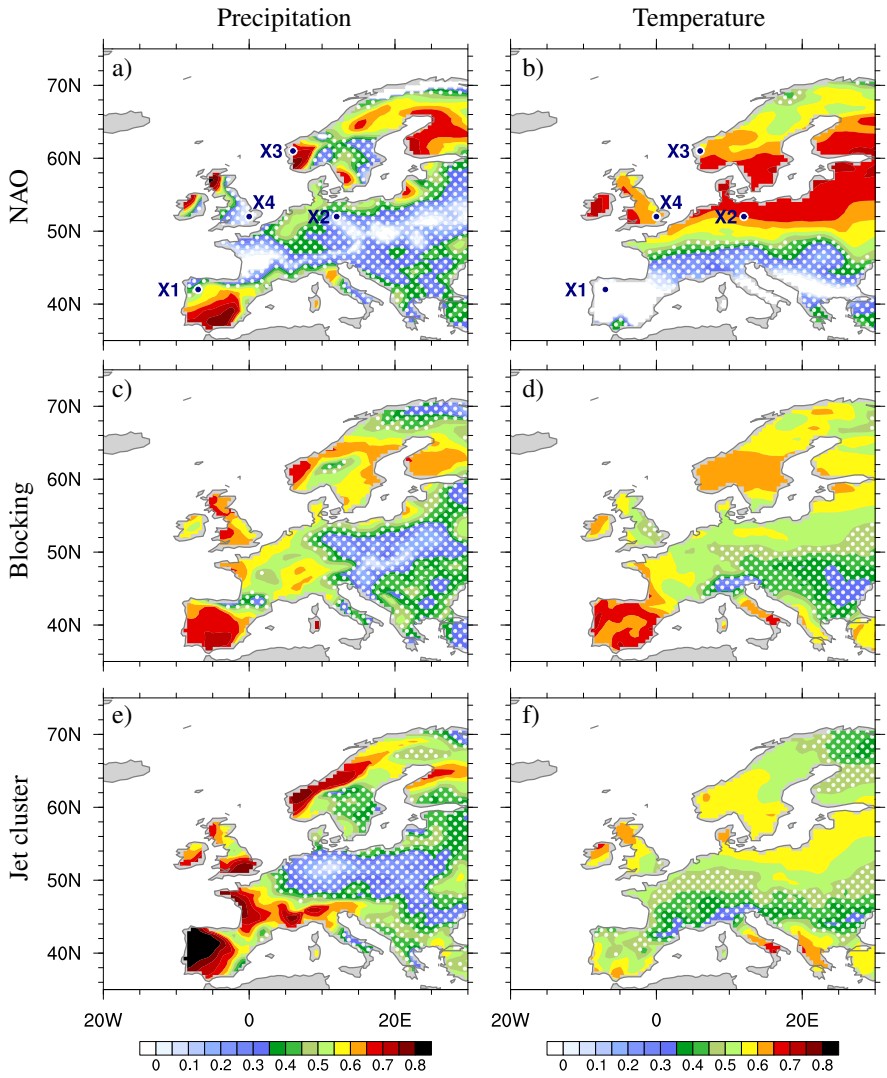

**Figure 6.** Correlation coefficient between seasonal anomalies of observed and reconstructed precipitation (left) and temperature (right) over Europe for two NAO phases (a-b), three blocking categories (c-d) and five jet clusters (e-f). White dots mark regions with correlation below 0.5. The blue dots labeled X1-4 in (a-b) indicate the four locations shown in Figure 10.

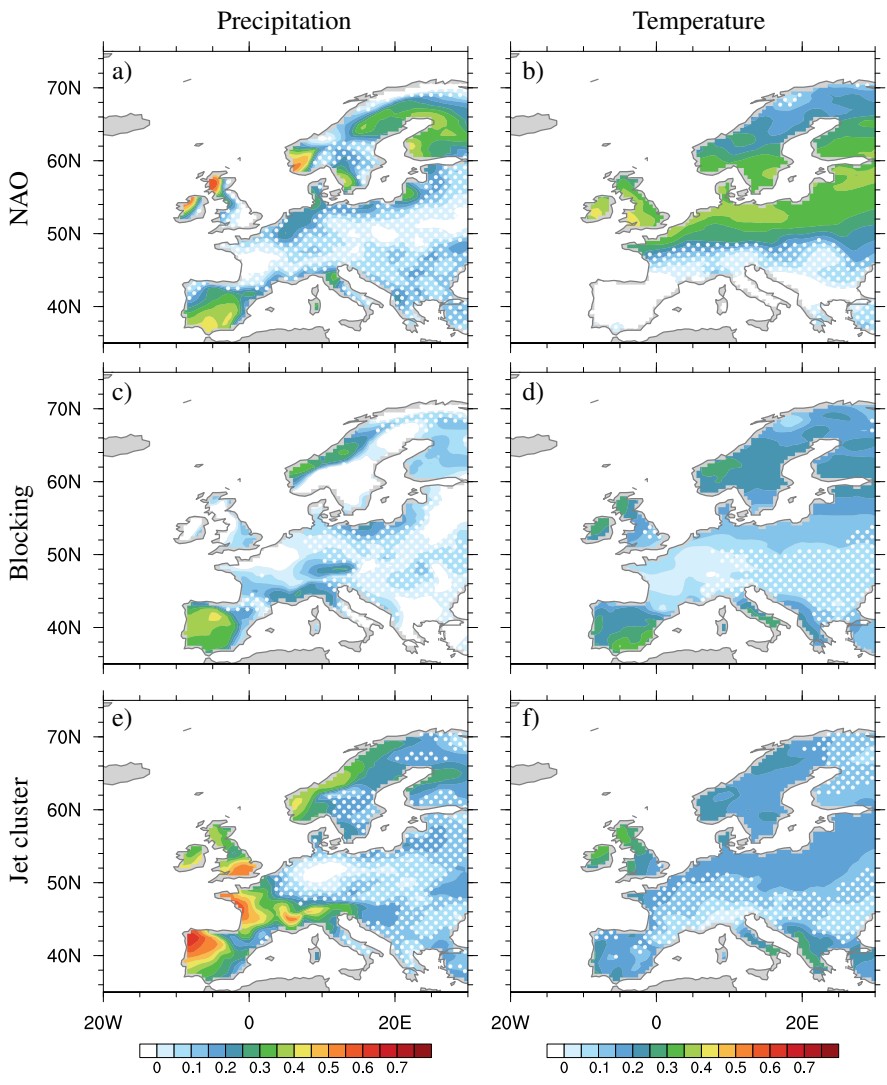

**Figure 7.** Coefficient of efficiency (CE) for Europe for precipitation (left) and temperature (right) for two NAO phases (a-b), there blocking categories (c-d), and five jet clusters (e-f). White dots mark regions with correlation below 0.5 (as in Figure 6).

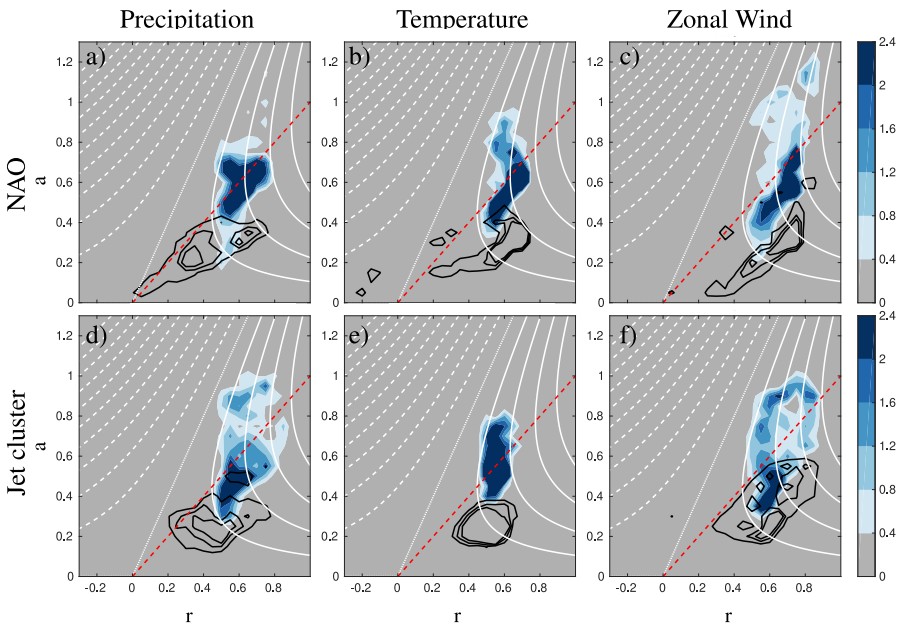

**Figure 8.** Frequency distribution of correlation ($r$) versus $a$, the ratio of the standard deviations of the reconstructed vs. observed seasonal time series (equation 6) over Europe (land only) for precipitation (first column), temperatures (second column) and zonal wind (third column) for two NAO phases (a-c), and the five jet clusters (d-f). Black contours (0.8, 1.6 and 2.4%) are for unscaled reconstructed time series, while the shading are the shifted (scaled) values applied only to grid points with correlation greater than 0.5. Data are normalized by the number of points in each distribution and units are given in percent. White lines are isolines of CE following equation (5) in 0.2 intervals (0 line dotted; positive solid, and negative dashed), while the red line shows the maximisation of CE as function of $r$ (i.e. the $r = a$ line).

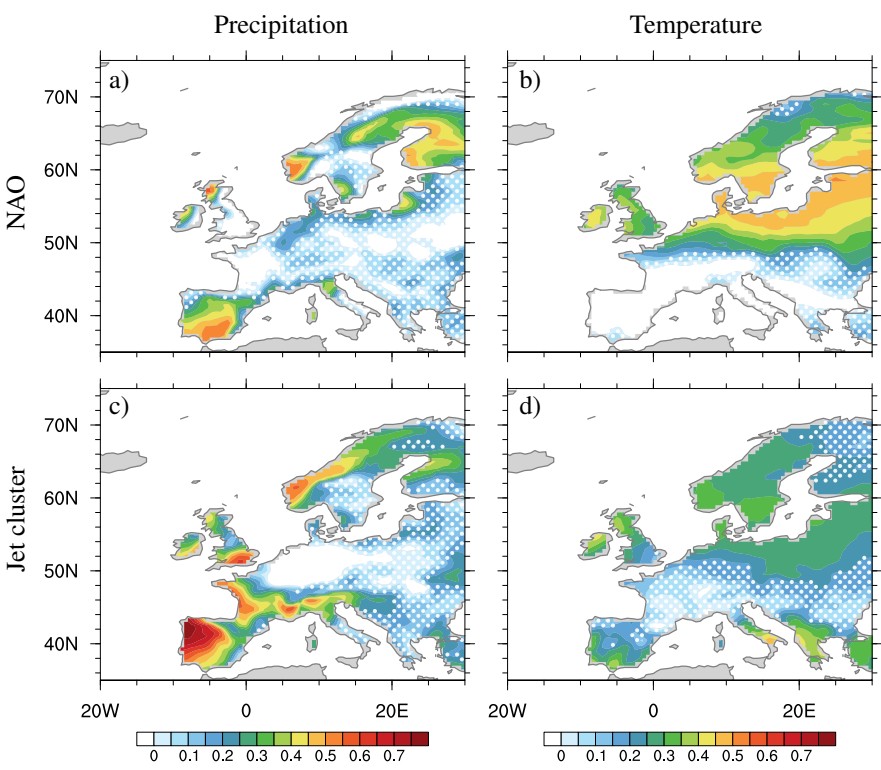

**Figure 9.** CE scaled reconstructions of precipitation (left) and temperature (right) for two NAO phases (a-b), and five jet clusters (c-d). Scaling factors are calculated using only points with correlation $r > 0.5$ (white dots marks regions with $r < 0.5$). The scaling coefficient are shown in Table 2.

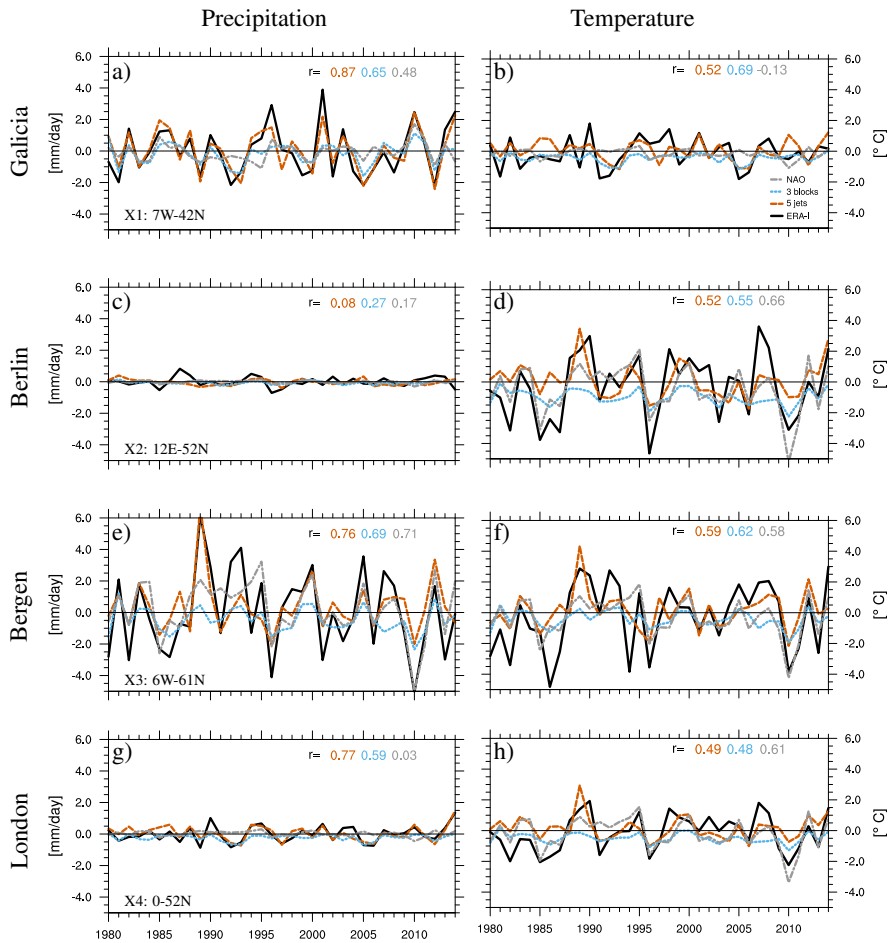

**Figure 10.** Time series of DJF precipitation (left, in mm day$^{-1}$, averaged over a season) and temperatures anomalies (right, in °C) for four locations shown in Figure 6: Galicia (X1, a-b), Berlin (X2, c-d), Bergen (X3 e-f), and London (X4, g-h). The magnitude of the anomalies has been multiplied by the scaling factor $\beta$ for the NAO and jet, but not for the blocking (see main text). Correlation ($r$) between the time series of observed (ERA-I) and reconstructed anomalies for the different methods are shown using the colour legend in (b). Please note that the multiplication by the scaling factor has no effect on correlations. The same y-scale is used in each panel to highlight the large differences in magnitude and variability across the different locations.

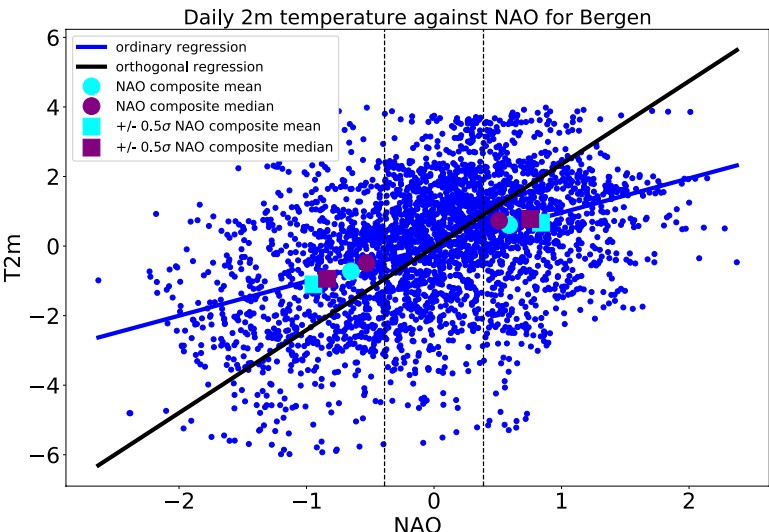

**Figure 11.** Illustration of the different regression slopes from ordinary (blue) and weighted orthogonal (black) regression on daily values of 2m temperature (T2m) the grid-box closest to Bergen, Norway, plotted against NAO. The cyan (purple) circles and squares show composite mean (median) values for all positive/negative NAO days and all positive/negative NAO days with |NAO| > 0.5σ.