# Peer review of "Reconstructing winter climate anomalies in the Euro-Atlantic sector using circulation patterns"

_Weather and Climate Dynamics, 2021_

## Referee Comment (RC1)

**1 General comments**

The authors present a really nice study about the reconstruction of winter climate anomalies in the Euro-Atlantic sector using different measures to classify large scale flow anomalies. This is done with the broadly used global pattern index of the NAO, widely recognised as an important factor to influence winter anomalies over Europe. The authers further use blocking regions and jet classifications in the North Atlantic to investigate how well European winter anomalies, in temperature and precipitation, can be reconstructed by those groups (NAO, blocking jet). The author do not do this solely by using simple correlations, but also another measure also accounting for the represented temporal variability (amplitude) of those groups. The manuscript is very well structured, everything is quite well explained and results are presented in a very coherent way, guiding the reader through the individual steps of the results and how they are connected. The authors also spent enough time to explain the methods and give the reader the necessary information (e.g. including Fig. 1) to understand the applied diagnostics without having to check the references for the essential informations. All in all, very nicely done.

I only have two major issues. The smaller issue concerns the application of the diagnostics in the result section. It wasn't always fully clear which of the specific representation of their diagnostics is applied, e.g. concerning the CE representation. While reading the result section I thought the authors always use the scaled reconstruction (with $\beta$), only after they explicitly mentioned that they introduce now the scaling did it become clear to me that they didn't use it before. Also, the authors could have clarified how Equ. (4) and Equ. (6) are used in this manuscript. If I'm not mistaken, only the simplification (Equ. 4) is used. And although highlighting the different Equations, the authors did not make a statement on the impact of the different choices, except for saying that the assumption of a zero climatological mean of the reconstructed anomalies should be fullfilled. More specific comments about this can be found in the "Specific comments" section.

My main issue with the manuscript is that the authors maybe did not the best job in highlighting the importance of their study and the usefulness of their presented results. The final discussion is full of very interesting aspects which are closely linked to this study, but which weren't analysed. This includes for example the extension of the regions to identify the classifications of a specific group (e.g. the limited eastward extension of the jet-region), the number of classifications within a group, the impact of extreme events for this kind of climate reconstruction. Also, the main conclusion I did get from this is that somehow all those groups are able to some extent to reconstruct climate anomalies, but it is also not clear which one is best. The thing is that it cannot be surprising that any kind of large scale feature which represents a large part of the variability of winter anomalies (variability in the same variable) is very likely able to identify large scale seasonal anomalies in other variables like temperature or precipitation, as those variables are linked to each other (no rain within a blocking high, storms moving along the jet, etc.). I think the authors could made clearer what there main message is, why it might be important to use different classes. How can those results

interpreted and maybe applied to other fields (e.g. if climate models are able to capture the general jet behaviour over the Atlantic this might be not enough as the exact representation of the established blocking periods decides on whether or not the model can have skill for specific parts of Europe in reconstructing the climate). Not sure the above examples make any sense at all, but I try to say with that that the authors could put some more effort into selling their results, why is this study so useful (I think this manuscript is slightly missing this). This is probably even more important because the authors restricted their analysis strongly to the choices they have done. As previously mentioned, and also raised by the authors, there are so many interesting modifications of the setup of the study and modifications of the choices which weren't done by the authors, although probably having important impact on the results. Also are there a few questions arising from the conclusions of the authors, e.g. can the authors quantify the relevance of remote regions (Pacific jet configurations, ENSO, North American or Pacific blocking) for the European winter climate reconstruction (as authors highlight the importance of jet classification region to European climate response). To summarize, I think if the authors could highlight more the important aspects of their results beyond the connection that large scale flow anomalies are connected with large scale climate response, the impact and relevance of this study could be improved.

One example for highlighting the results of this manuscript could be the connection between the different groups of large scale flow representation. Not sure how new and innovative this is, but I really appreciated how well the authors presented the connection between those large scale flow anomalies (link between NAO, blocking and jet). This was really nicely presented (e.g. Fig 2-4 and Fig. 5). Not sure if this was already done in a similar fashion in other studies, but I expected this to be highlighted at the end, because it nicely show the connections and links between those different groups (not only using only one group, like NAO, but compare it with different large scale flow realisations and show how those are connected). Maybe one could have also interpreted how this can be used to understand the similar usefulness to reconstruct seasonal anomalies or how specific aspects/regions cannot be as well reconstructed as others for a specific group.

**2 Specific comments**

**2.1 Introduction**

**lines 29-30:**
I wonder if "obstructs" is the appropriate word, as specific locations of a blocking high can also lead to a northward or southward shift of the jet - so changing the path of the jet but not really fully blocking it. So maybe it would be better to specify "obstruction" in terms of obstructing the climatological position of the westerly flow (which of course will have an impact on seasonal anomalies in temperature, etc.)?

**line 36:**
Not sure how well the term storminess is established. I think it would need a short explanation. Therefore I would suggest to remove "with distinct patterns of storminess", because the sentence works also without this term.

**Table 1:**
Maybe replace "NAO values above (below) 0.5" with "NAO values above 0.5 (below -0.5)".

**lines 83-87:**
Reading this, one might wonder about the choice of the distance $d = 0.5$ if $46\%$ cannot be classified. This does sound like only the days very close to a centroid are chosen, which seems fine to create clear jet classifications. However, the explanation is somewhat surprising, namely that the large group of undefined cases is in particular occuring during the transition days, which would mean that either the the different jet classification are associated with frequent changes or that the transition phase between the jet classifications is rather long. Is the choice of d=0.5 the same as in Madonna et al (2017)?
**reference also to section 2.2.2:**
Also having read the part of A_rec, this jet classification seems relevant, as stricter classification choices (larger d) leads to less days recognized as jet classification. Since A_rec is normalized by the full number of days (90), this could lead to an increase in the amplitude error (for CE, Equ. 4). So introducing the beta factor (Equ. 7) is essential to account for this and therefore corrects possible "missing" members of specific classifications, or am I mistaken? But in lines 137-139 the authors only mention the problem of a non-gaussian distribution which "requires" a correction with this $\beta$-factor. Not sure I made myself clear, therefore I include a short example. If I could assume that the seasonal anomalies are mainly driven by a N-jet and a S-jet, the choice of a larger d can lead to a underrepresentation of the anomalies for seasons mainly associated with either a N-jet or S-jet, meaning that the associated a-value will become smaller and with it will affect CE. So the $\beta$-factor will boost the dominant anomaly. If this is true, maybe this should also be included as a reason in lines 137-139.

**lines 95-97:**
Why do the authors not apply the same anomaly calculation to all variables? The position of the jet does also follow a seasonal cycle, following the location of the temperature gradients. Therefore, what is the reason for not applying the same anomaly calculation? If there is a reason for doing so, e.g. signal otherwise too noisy because of the mentioned strong fluctuations, I think this is worth mentioning. Otherwise the reader will wonder about the different procedures for the different variables.

**Equation (3):**
This refers to Equation (3) and the whole section 2.2.1. According to Equ. (3), CE represents a 2d field and the given equations and text (CE values of 0, 1, etc) are for time series at one gridpoint, correct? While first scanning the manuscript, I expected CE to be a single-value metric to capture the overall skill of the 2d reconstruction (because

the examples were for one value, e.g. $CE = 1$), but in section 2.2.1 references are only made to time series, so CE must have values at each longitude and latitude. To make this fully clear and prevent confusion, CE could be written in the same way as A_rec, as $CE(\phi, \lambda)$. Alternatively a short sentence mentioning this would be helpful.

**Section 2.2 Seasonal reconstructions:**
The underlying assumption of this procedure is that the unclassified days more or less represent the climatological mean, otherwise the large undefined classifications (with representing partly more than 50 %) would account for large anomalies that this procedure will never be able to capture. This means for normally distributed quantities, as long as there is no mixing of different classifications (e.g. one jet classification and one blocking classification), that this method tests how well the undefined classifications represent the climatology (i.e. is there a missing classification with strongly impacts seasonal anomalies), right? Maybe this is worth mentioning (if true), because this could get lost in the text with all the description of weighting procedure and combination of classifications? However, non-normally distributed quantities would also lower the reconstruction skill. So probably these two aspects can lead to a reduced reconstruction skill, non-climatological rest-classifications and non-normal distributions. Maybe worth mentioning here, because both aspects are a necessary assumption for this method.

**line 120:**
Same as in last comment, the assumption hereby is that the undefined classifications are close to climatology, otherwise one cannot assume that the average of p is zero. As this is now part of the definition of CE (Equation 4), I think this should be clarified, or alternatively the step from Equ. (3) to Equ. (4) should be justified.

**line 129:**
Where does the CE= 0.25 comes from? Is this a usual value indicating good reconstructions, also used in other studies? Or alternatively the authors could give some further insight into the simple function CE. The global maximum of this CE function, as function of a, is at the location r=a, which means that the maximum value of CE for any a is r**2. This would mean that independent of the amplitude error (or a), the choice of CE= 0.25 requires a minimum correlation of 0.5, whereas for a perfect correlation the values of a must be between 0.13 and 1.87 (if I am not mistaken). I would find such a simple consideration of CE very helpful, otherwise I would feel a bit lost when reading about the choice of 0.25 (because initially, I had no feeling what this value means in terms of correlation or amplitude error). What I find slightly confusing about this CE-value is that for exceeding the 0.25 value at low correlations (slightly above 0.5), it must include an amplitude error ($a < 1$) in order to exceed the threshhold of 0.25, so low correlations can be corrected by a reduced variance in the reconstructed fields. But the overall conclusion from this function is, that it is a well established, easy measure to evaluate the reconstruction of an original field?

**lines 139-140:**

But using the median would lead to a bias in the overall climatological reconstruction of this variable (sum/average over all seasons), wouldn't it? But this would be fine, because this bias is not relevant for the calculation of CE, is this correct? So what I found surprising while first reading this sentence is that there is no condition that the climatological reconstruction of this method does not necessarily need to represent the climatology. Sorry, not sure this even is a question, or if it is obvious and not worth mentioning.

**line 145:**
"...calculate the centre of mass of the CE...". Is there a weighting by the value of CE involved to calculate the centre of mass, or is this purely by distance in correlation-amplitude space with each value represented by a weighting value of 1 (e.g. what is the "mass" here)? I see, in line 149 the points are restricted to correlation values of above 0.5, which probably suggests that the weights are all 1. I would suggest to be clarify this in the text.

**line 155:**
Is uncertainty the correct term here? It does not really test the uncertainty of the calculation of CE, but checks if the resulting CE-values are statistical significant, doesn't it?

**2.2 Results:**

**line 190, Fig. 5:**
Very nice overview of the seasonal connection between different, already coherently grouped, classifications. It would be nice if the NAO- would also have the associated correlation value with the jet classification, as all the other values have. Now with the missing r-value for NAO-, one has to check always the figure caption to understand for which variable GB has a correlation of 0.64. I would also include the NAO+ correlations in the legend. As the variable without a correlation value is the one with which the correlation is calculated, there is never any confusion.

**lines 190-197:**
Coming back again to the choice of the d-value for the jet classifications. I think it would be helpful if the authors explicitly mention their "large" choice of $d = 0.5$. As for all cluster groups (e.g. NAO-, GB and S-jet - IWB and N-jet - etc) the jet part represents the smallest part (in terms of number of days), it seems like this is due to the restrictive choice of d and a smaller d would bring the number of days for the jet-groups closer to the rest. I would suggest the authors, as already previously mentioned, give some further support for this choice. If a smaller d than 0.5 is problematic, is this really due to the transition periods as the text of the authors seems to indicate (line 85)? If this is the case, maybe it is worth mentioning more specifically what the problem about the transition period is (are those so frequently that they can make up nearly 50 percent of the time?). If $d = 0.5$ was also chosen in the original paper and discussed in depth, I would suggest to explicitly cite it in relation to this aspect.

**lines 200-201:**
I see, very nice. Maybe it is worth already mentioning this while introducing the reconstruction method. However, this is for the composite of the full climaotology. From what I mentioned before, my assumption was that for individual seasons with dominance of specific classifications (like N-jet), the anomaly of the unclassified cases will show a similar pattern. To estimate this effect a comparison of the variance would be helpful to get a deeper insight. Having a very small value of the anomalies for the climatological composite is nice to conclude, that the remaining unclassified cases indeed represents climatology, but it would also be nice to have a similar short comment about the variance. If it is small then the reader knows that there is absolutely nothing to worry about the procedure in this manuscript. If it is not really small, the authors could give a short explanation why they think that is (no otpimal way of defining the classifications because of some cutoff problem between getting the relevant patterns and reducing the variance of the unclassified cases?).

**lines 235-236, Figure S2:**
Oh, it seems I did not understand the uncertainty estimation correctly. I thought that this resampling of the 35 years is done to either observation or reconstruction to get an estimate of how significant the resulting CE-value is. But in this case, I would assume that for the areas of very high correlations, some of them would also show very different values (lower) for the bootstrap mean, because the reconstructed time series was already capturing well the observations, whereas a resampling of the years will not do so. So I'm slightly puzzled about the near identity between CE and bootstrap mean, suggesting that I don't understand what wasactually done. Maybe the authors can therefore explain in a bit more detail how the uncertainty estimation does or what it is for?

**line 262:**
That wasn't clear to me. I thought the scaling factor was always used after being introduced in the method section. Not sure I have missed this while reading through the text, but I would suggest to make this clear while showing CE for the first time (if not done and I just missed that).

**lines 274-275, Fig. 10:**
I would suggest to include the 0-line in the anomaly plots, which makes the reading of those plots much easier. This would also highlight the issue of the blocking classifications, that this one does for some regions seems to be only able to capture negative anomalies (10b, d and h). Further, it seems a bit confusing when talking about amplitudes while interpreting this plot. In my mind I quickly switched to my usual interpretation of amplitude, representing the strength of the anomaly. This was confusing, because e.g. Fig. 10b and d seem to indicate that for the blocking reconstruction negative amplitudes of the anomalies are being overestimated while not being able to represent positive anomalies. But here amplitude refers to a from the CE calculation, right? I would suggest to mention this again explicitly, which makes following the interpretations easier.

**2.3 Discussion and Conclusion**

**line 286:**

Although, the better reconstruction is more focused towards western Europe, whereas temperature is better reconstructed at more eastern and southeastern Europe (Fig. 9). So what does in general mean, a higher mean CE value, occurrence of very high CE values or more points showing $CE > 0.25$? I think the authors could present this a lot more convincingly, if they could attribute a quantification to it, using some sort of simple measure as mentioned above. Just by looking again at Fig. 9 this doesn't seem to be so obvious. I appreciate the authors give some reasoning behind the differences between temperature and precipitation in this paragraph, which represents some explanation beyond just showing the results of this correlation/CE test.

**line 305:**

So what is the actual main message of the manuscript. As long as one choses classifications which are associated with large scale flow anomalies (NAO, blocking, jet), one will be able to capture to some extent the seasonal large scale anomalies over Europe. But this study does not come to a conclusion what metric is best and it also depends on what someone might be interested in (precipitation or temperature). So the main message is that there is no perfect metric to capture both temperature and anomalies? A possible limitation of the reconstruction could be due to extreme events, but those are not considered in this study.

**lines 298-304:**

I think I made t his point already earlier: isn't this (necessity of including a scaling factor) also a problem of the reconstruction method? If one would only use NAO values above 2 or below -2, it would be obvious that the reconstruction would be underestimate the amplitude, wouldn't it? So going down with this threshold to 1 would probably do a better job in reconstructing the seasonal anomalies, because more seasons will be represented within this class and more variability is possible. Whereas defining a class of NAO with values above -5 would do a very bad job, because it will only reproduce climatology. So there is some sort of useful cutoff, or the classification are already defined in a useful way (by EOF, clustering, etc are they already constructed in a way to capture the variability). So I assume my question is, if this shouldn't be rather discussed with more focus on the method / discussion of how necessary a scaling is to start with?
Further, I think it was not fully clear in the manuscript which definition of CE was used (I raised this point also in another comment). It wasn't clear to me that the first results were for the non-scaled CE values (sorry if I just missed it). But also the choice of CE (Equ. 4 or Equ. 6) was not fully clear to me. If I am not mistaken, Equ. (4) was taken to calculate the CE values, right? Was Equ. (6) also used somewhere? In general, what is the reason to use Equ. (4)? Since Equ. (6) doesn't really need more calculations or is too complicated, it is not obvious for me why exactly the simplification was used? Is the reason behind this the identification of a useful scaling (Fig. 8)? I think the authors could make this a bit clearer. Also, I hope I didn't miss it, but did the authors compare

the two calculations, how different are the results? This information could be useful to justify the general use and handling of Equ. (4).

**lines 305-314 and lines 315-324:**
Really interesting aspect in those two paragraphs. But the comparison between NAO (with large region) and jet (with smaller region) is difficult, because of different underlying variables. Further, a larger region does not necessarily help to get better correlations, as large part of the NAO region is also far way from Europe. Therefore, the argument of the NAO region has the advantage to capture also the European region is not really convincing, as the NAO, as mentioned by the authors, also only includes 2 classifications for the whole region. Apart from this, the jet-region also includes part of western Europe, and as Europe is part of the westwind regime, it is not really surprising that the upstream information has an impact on the immediate downstream weather response. Extending the jet-region eastward would be very interesting to see, if this indeed does substantially increase the reconstruction skill, but I would assume this is not clear. So this point is very interesting, but it was not investigated here. Same is true for the next interesting point (number of classifications for one variable), also very interesting but also not included in this manuscript. The question about the relevant or optimal number of classes for the reconstruction exercise does sound also really interesting. But it is not clear to me, what the example of the 30 MSL classifications tells the reader, as at some point maybe too many classifications could be reducing the skill in a similar way as extreme events could be problematic. The more classifications, the more relevant individual extremes could become, because their impact will be felt more strongly during individual seasons.

**lines 325-330:**
Same as for lines 305-314 and 315-324. The authors nicely present a way of how their results can be interpreted, where relevant impacts of the method on the results might occur or how specific changes in the classification setup could have an impact. I find this discussion very helpful, but I am somehow missing the part where the authors are highlighting the results of their study. Main issue with this point can be found in my general comment.

---

## Author Comment (AC1)

**Revision of wcd-2021-6 "Reconstructing winter climate anomalies in the Euro-Atlantic sector using circulation patterns" by Madonna et al.**

We would like to thank the reviewers for their constructive comments and suggestions.
In response to the main issues raised by the reviewer, we plan to:
- provide a sensitivity analysis for the choice of the thresholds (e.g. d)
- clarify how the CE has been calculated, discuss the assumptions and their validity for our calculation, and improve the presentation and application of the scaling factor.
- better highlight the importance of the study and its implications

Reviewers' comments are in black, our replies are in blue.

**Reviewer 1**

1 General comments

The authors present a really nice study about the reconstruction of winter climate anomalies in the Euro-Atlantic sector using different measures to classify large scale flow anomalies. This is done with the broadly used global pattern index of the NAO,widely recognised as an important factor to influence winter anomalies over Europe.The authors further use blocking regions and jet classifications in the North Atlanticto investigate how well European winter anomalies, in temperature and precipitation,can be reconstructed by those groups (NAO, blocking jet). The author do not do this solely by using simple correlations, but also another measure also accounting for the represented temporal variability (amplitude) of those groups. The manuscript is very well structured, everything is quite well explained and results are presented in a very coherent way, guiding the reader through the individual steps of the results and how they are connected. The authors also spent enough time to explain the methods and give the reader the necessary information (e.g. including Fig. 1) to understand the applied diagnostics without having to check the references for the essential informations. All in all, very nicely done.I only have two major issues. The smaller issue concerns the application of the diagnostics in the result section. It wasn't always fully clear which of the specific representation of their diagnostics is applied, e.g. concerning the CE representation. While reading the result section I thought the authors always use the scaled reconstruction (with β), only after they explicitly mentioned that they introduce now the scaling did it become clear to me that they didn't use it before. Also, the authors could have clarified how Equ. (4) and Equ. (6) are used in this manuscript. If I'm not mistaken, only the simplification (Equ. 4) is used. And although highlighting the different Equations, the authors did not make a statement on the impact of the different choices, except for saying that the assumption of a zero climatological mean of the reconstructed anomalies should be fullfilled. More specific comments about this can be found in the "Specific comments" section.

We agree that in section 2.2.2, the description of how the diagnostic is used is not totally clear. Linked to the comment of reviewer 2, we will move the introduction of the scaling factor to section 3.3, where we use it. Also, we will clarify in the text that we first compute the coefficient of efficiency using raw data, and only afterwards do we use the scaling factor. We also noticed that we were not clear on which formula was used to calculate the CE - we use the full equation (3), and the approximation (4) was used only to determine the scaling factor. We will make sure that in the revised version we better describe the procedure and reference better to the equations.

My main issue with the manuscript is that the authors maybe did not the best job in highlighting the importance of their study and the usefulness of their presented results. The final discussion is full of

very interesting aspects which are closely linked to this study, but which weren't analysed. This includes for example the extension of the regions to identify the classifications of a specific group (e.g. the limited eastward extension of the jet-region), the number of classifications within a group, the impact of extreme events for this kind of climate reconstruction. Also, the main conclusion I did get from this is that somehow all those groups are able to some extent to reconstruct climate anomalies, but it is also not clear which one is best. The thing is that it cannot be surprising that any kind of large scale feature which represents a large part of the variability of winter anomalies (variability in the same variable) is very likely able to identify large scale seasonal anomalies in other variables like temperature or precipitation, as those variables are linked to each other (no rain within a blocking high, storms moving along the jet, etc.). I think the authors could made clearer what there main message is, why it might be important to use different classes. How can those results interpreted and maybe applied to other fields (e.g. if climate models are able to capture the general jet behaviour over the Atlantic this might be not enough as the exact representation of the established blocking periods decides on whether or not the model can have skill for specific parts of Europe in reconstructing the climate). Not sure the above examples make any sense at all, but I try to say with that that the authors could put some more effort into selling their results, why is this study so useful (I think this manuscript is slightly missing this). This is probably even more important because the authors restricted their analysis strongly to the choices they have done. As previously mentioned, and also raised by the authors, there are so many interesting modifications of the setup of the study and modifications of the choices which weren't done by the authors, although probably having important impact on the results. Also are there a few questions arising from the conclusions of the authors, e.g. can the authors quantify the relevance of remote regions (Pacific jet configurations, ENSO, North American or Pacific blocking) for the European winter climate reconstruction (as authors highlight the importance of jet classification region to European climate response). To summarize, I think if the authors could highlight more the important aspects of their results beyond the connection that large scale flow anomalies are connected with large scale climate response, the impact and relevance of this study could be improved. One example for highlighting the results of this manuscript could be the connection between the different groups of large scale flow representation. Not sure how new and innovative this is, but I really appreciated how well the authors presented the connection between those large scale flow anomalies (link between NAO, blocking and jet). This was really nicely presented (e.g. Fig 2-4 and Fig. 5). Not sure if this was already done in a similar fashion in other studies, but I expected this to be highlighted at the end, because it nicely show the connections and links between those different groups (not only using only one group, like NAO, but compare it with different large scale flow realisations and show how those are connected). Maybe one could have also interpreted how this can be used to understand the similar usefulness to reconstruct seasonal anomalies or how specific aspects/regions cannot be as well reconstructed as others for a specific group.

We thank the reviewer for the thoughtful comments and suggestions. We are happy to expand the discussion and conclusion to better highlight the importance of the study and its implications. Building from the reviewer's line of thought, the study could help to understand, in climate change scenarios, what types of weather events contribute to changes in the climatological temperature and precipitation. It could also be useful for extended range predictability, as for example the S-jet regime (Madonna et al. 2019) and the NAO- (Cassou 2008) have been shown to occur preferentially after specific MJO phases that are influenced by SST in the equatorial Pacific. In Madonna et al 2017, we reconciled the characterisation of the atmospheric variability using jet clusters, weather regimes and the NAO index. However, we did not directly compare the different frameworks from the point of view of climate impacts. The present study would be a good chance to do so.
We will also expand our discussion on the sensitivity of the results to some of the analysis choices we made, as suggested by the reviewer (such as the d threshold).

2 Specific comments

2.1 Introduction

lines 29-30: I wonder if "obstructs" is the appropriate word, as specific locations of a blocking high can also lead to a northward or southward shift of the jet - so changing the path of the jet but not really fully blocking it. So maybe it would be better to specify "obstruction"in terms of obstructing the climatological position of the westerly flow (which of course will have an impact on seasonal anomalies in temperature, etc.

We will reformulate the sentence to acknowledge the possibility of a flow deviation. "In contrast, blocking is the presence of a persistent and stationary high-pressure system that obstructs or deviates the westerly flow."

line 36: Not sure how well the term storminess is established. I think it would need a short explanation. Therefore I would suggest to remove "with distinct patterns of storminess", because the sentence works also without this term.

We will replace "storminess" with "patterns of storm propagation,".

Table 1: Maybe replace "NAO values above (below) 0.5" with "NAO values above 0.5 (below-0.5)".

Thank you for the suggestion that we will implement.

lines 83-87: Reading this, one might wonder about the choice of the distance d= 0.5 if 46 % cannot be classified. This does sound like only the days very close to a centroid are chosen, which seems fine to create clear jet classifications. However, the explanation is somewhat surprising, namely that the large group of undefined cases is in particular occuring during the transition days, which would mean that either the different jet classification are associated with frequent changes or that the transition phase between the jet classifications is rather long. Is the choice of d=0.5 the same as in Madonna et al(2017)?

In Madonna et al 2017 all days are used, as the purpose of the study was to reconcile the weather regime and jet variability frameworks. It is not exactly that 46% of the days can not be classified, but that 46% of the days share similarities with more than one cluster. This often happens during transitions days, as illustrated in the table below. Previous studies also showed that transitions from a jet state to another are not instantaneous (e.g. Franzke et al 2011) and occur relatively frequently (~ median persistence of a jet regime is ~ 1 week, e.g. Franzke et al 2011, Madonna 2017). However, not all the unclassified days are transition days. For example, extreme events that do not resemble any of the cluster can belong to the unclassified category.

As an example, we show the inverse distance (1/d) from the centroid for 1981.01.14 to 1981.01.21:

| 1/d | N | M | T | C | S |
|-----|-----|-----|-----|-----|-----|
| 1981.01.14 | 0.79 | 0.04 | 0.05 | 0.08 | 0.03 |
| 1981.01.15 | 0.61 | 0.06 | 0.09 | 0.17 | 0.06 |
| 1981.01.16 | 0.49 | 0.07 | 0.12 | 0.23 | 0.09 |
| 1981.01.17 | 0.43 | 0.07 | 0.15 | 0.23 | 0.12 |
| 1981.01.18 | 0.42 | 0.07 | 0.19 | 0.19 | 0.13 |

| | | | | | |
|---|---|---|---|---|---|
| 1981.01.19 | 0.44 | 0.08 | 0.22 | 0.14 | 0.13 |
| 1981.01.20 | 0.46 | 0.08 | 0.26 | 0.09 | 0.11 |
| 1981.01.21 | 0.47 | 0.08 | 0.28 | 0.07 | 0.1 |
| 1981.01.22 | 0.46 | 0.09 | 0.31 | 0.05 | 0.09 |
| 1981.01.23 | 0.4 | 0.1 | 0.37 | 0.04 | 0.08 |
| 1981.01.24 | 0.26 | 0.13 | 0.5 | 0.04 | 0.08 |
| 1981.01.25 | 0.12 | 0.15 | 0.63 | 0.03 | 0.07 |
| 1981.01.26 | 0.07 | 0.19 | 0.65 | 0.03 | 0.07 |

This period includes a transition from a northern (N) jet to a tilted (T) jet. Some of the transition days are quite similar to more than one centroid, while others have a d < 0.5 and are counted as undefined days.

We agree that the choice of d=0.5 is rather restrictive but was introduced to be sure to capture only days that clearly belong to a certain cluster. Linked to the major comment of reviewer 1 and also a comment from reviewer 2, we will perform a sensitivity analysis to the choice of d.

Franzke, C., Woollings, T., & Martius, O. (2011). Persistent circulation regimes and preferred regime transitions in the North Atlantic. *Journal of the atmospheric sciences*, *68*(12), 2809-2825.

reference also to section 2.2.2:
Also having read the part of Arec, this jet classification seems relevant, as stricter classification choices (larger d) leads to less days recognized as jet classification. Since Arec is normalized by the full number of days (90), this could lead to an increase in the amplitude error (for CE, Equ. 4). So introducing the beta factor (Equ. 7) is essential to account for this and therefore corrects possible "missing" members of specific classifications, or am I mistaken? But in lines 137-139 the authors only mention the problem of a non-gaussian distribution which "requires" a correction with this β-factor. Not sure I made myself clear, therefore I include a short example. If I could assume that the seasonal anomalies are mainly driven by a N-jet and a S-jet, the choice of a larger d can lead to a underrepresentation of the anomalies for seasons mainly associated with either a N-jet or S-jet, meaning that the associated a-value will become smaller and with it will affect CE. So the β-factor will boost the dominant anomaly. If this is true, maybe this should also be included as a reason in lines 137-139.

We thank the reviewer for the comment. As stated above, we plan to test the sensitivity of the anomalies to the choice of d. For example, we can test how the β-factor changes if all days are used for the reconstruction. The results of this will be included in the revision.

lines 95-97:Why do the authors not apply the same anomaly calculation to all variables? The position of the jet does also follow a seasonal cycle, following the location of the temperature gradients. Therefore, what is the reason for not applying the same anomaly calculation? If there is a reason for doing so, e.g. signal otherwise too noisy because of the mentioned strong fluctuations, I think this is

worth mentioning. Otherwise the reader will wonder about the different procedures for the different variables.

We agree with the reviewer that we could have removed the daily climatology of all variables. The choice to leave in the climatology was mainly because previous studies have shown that the seasonal cycle of wind (see Woollings et al 2014, their Figure 3a) is small within the DJF winter considered here. Also most studies looking at seasonality in precipitation aggregate DJF (e.g. Zveryaev 2004 or Rauthe et al 2013). We will mention this in the manuscript.

Woollings, T., Czuchnicki, C., & Franzke, C. (2014). Twentieth century North Atlantic jet variability. *Quarterly Journal of the Royal Meteorological Society*, *140*(680), 783-791.

Zveryaev, I. I. (2004), Seasonality in precipitation variability over Europe, *J. Geophys. Res.*, 109, D05103, doi:10.1029/2003JD003668.

Rauthe, M., Steiner, H., Riediger, U., Mazurkiewicz, A., & Gratzki, A. (2013). A Central European precipitation climatology–Part I: Generation and validation of a high-resolution gridded daily data set (HYRAS). *Meteorol. Z*, 22(3), 235-256.

Equation (3): This refers to Equation (3) and the whole section 2.2.1. According to Equ. (3), CE represents a 2d field and the given equations and text (CE values of 0, 1, etc) are for timeseries at one gridpoint, correct? While first scanning the manuscript, I expected CE tobe a single-value metric to capture the overall skill of the 2d reconstruction (because the examples were for one value, e.g.CE= 1), but in section 2.2.1 references are only made to time series, so CE must have values at each longitude and latitude. To make this fully clear and prevent confusion, CE could be written in the same way as Arec, as $CE(\varphi, \lambda)$. Alternatively a short sentence mentioning this would be helpful.

The reviewer is right, CE is calculated at each grid point. In the revised version of the manuscript, we will clarify this point and modify the equation according.

Section 2.2 Seasonal reconstructions: The underlying assumption of this procedure is that the unclassified days more or less represent the climatological mean, otherwise the large undefined classifications (with representing partly more than 50 %) would account for large anomalies that this procedure will never be able to capture. This means for normally distributed quantities, as long as there is no mixing of different classifications (e.g. one jet classification and one blocking classification), that this method tests how well the undefined classifications represent the climatology (i.e. is there a missing classification with strongly impacts seasonal anomalies), right? Maybe this is worth mentioning (if true), because this could get lost in the text with all the description of weighting procedure and combination of classifications? However, non-normally distributed quantities would also lower the reconstruction skill. So probably these two aspects can lead to a reduced reconstruction skill, non-climatological rest-classifications and non-normal distributions. Maybe worth mentioning here, because both aspects are a necessary assumption for this method.

This is correct: the anomalies of the undefined days are almost zero, which means that the undefined category represents the climatology. We briefly mention this later in the text (L201) but in the revision we will also mention it here and include the figures showing the anomalies for the undefined days (e.g. in the supplement).

line 120:Same as in last comment, the assumption hereby is that the undefined classifications are close to climatology, otherwise one cannot assume that the average of p is zero. As this is now part of the definition of CE (Equation 4), I think this should be clarified, or alternatively the step from Equ. (3) to Equ. (4) should be justified.

We will modify the text to include this point and show the anomalies of the undefined days. We also will better explain that we used eq. 3 to calculate CE, while the approximation (4) is used only to determine the scaling factor.

line 129:Where does the CE= 0.25 comes from? Is this a usual value indicating good reconstructions, also used in other studies? Or alternatively the authors could give some further insight into the simple function CE. The global maximum of this CE function, as function of a, is at the location r=a, which means that the maximum value of CE for any a is r**2. This would mean that independent of the amplitude error (or a), the choice of CE= 0.25 requires a minimum correlation of 0.5, whereas for a perfect correlation the values of a must be between 0.13 and 1.87 (if I am not mistaken). I would find such a simple consideration of CE very helpful, otherwise I would feel a bit lost when reading about the choice of 0.25 (because initially, I had no feeling what this value means in terms of correlation or amplitude error). What I find slightly confusing about this CE-value is that for exceeding the 0.25 value at low correlations (slightly above 0.5), it must include an amplitude error (a <1) in order to exceed the threshold of 0.25, so low correlations can be corrected by a reduced variance in the reconstructed fields. But the overall conclusion from this function is, that it is a well established, easy measure to evaluate the reconstruction of an original field?

We thank the reviewer for this comment. We will complement the example of the interpretation of CE given in lines 125-129 with the very nice example illustrated by the reviewer. This will definitely help provide readers with more intuition about what the CE and our definition of a good reconstruction means. We have marked with white dots regions with low correlations (r<0.5) in Figure 7 and 9 and also used only the non-dotted regions to calculate the scaling factor. We are happy to add an additional comment to motivate this choice.

lines 139-140: But using the median would lead to a bias in the overall climatological reconstruction of this variable (sum/average over all seasons), wouldn't it? But this would be fine, because this bias is not relevant for the calculation of CE, is this correct? So what I found surprising while first reading this sentence is that there is no condition that the climatological reconstruction of this method does not necessarily need to represent the climatology. Sorry, not sure this even is a question, or if it is obvious and not worth mentioning.

The CE is calculated using the full equation (eq. 3 or 6), which takes into account the climatologies of the observed ($\bar{o}$) and reconstructed ($\bar{p}$) anomalies (eq. 6), thus accounting for possible non-zero $\bar{p}$.
In accordance with previous comments, we will clarify this part and state that the CE is calculated from the full time series and not using the approximation. This means that for the calculation of the CE it is not required that the climatological reconstruction is not biased. But biases in the reconstruction would play a role if we use the simplified formula (equation 4) to calculate the CE.

line 145:"...calculate the centre of mass of the CE...". Is there a weighting by the value of CE involved to calculate the centre of mass, or is this purely by distance in correlation-amplitude space with each value represented by a weighting value of 1 (e.g. what is the"mass" here)? I see, in line 149 the points are restricted to correlation values of above 0.5, which probably suggests that the weights are all 1. I would suggest to be clarify this in the text.

Yes, each grid point with r>0.5 has the same weight. We will clarify this in the text.

line 155: Is uncertainty the correct term here? It does not really test the uncertainty of the calculation of CE, but checks if the resulting CE-values are statistical significant, doesn't it?

With bootstrapping, we resampled the 35-year time series to assess the effect of internal variability on the CE. We show the standard deviation of the bootstrapped CE distribution, but we could instead have plotted a confidence interval (e.g., 5th to 95th percentile) as an alternative way to express the uncertainty (Wood 2004). A classical significant test would only test the null hypothesis, i.e. CE$\neq$ 0.

Wood, M. (2004). Statistical inference using bootstrap confidence intervals. Significance, 1(4), 180-182.

2.2 Results:
line 190, Fig. 5:Very nice overview of the seasonal connection between different, already coherently grouped, classifications. It would be nice if the NAO- would also have the associated correlation value with the jet classification, as all the other values have. Now with the missing r-value for NAO-, one has to check always the figure caption to understand for which variable GB has a correlation of 0.64. I would also include the NAO+ correlations in the legend. As the variable without a correlation value is the one with which the correlation is calculated, there is never any confusion. lines 190-197:Coming back again to the choice of the d-value for the jet classifications. I think it would be helpful if the authors explicitly mention their "large" choice of d= 0.5. As for all cluster groups (e.g. NAO-, GB and S-jet - IWB and N-jet - etc) the jet part represents the smallest part (in terms of number of days), it seems like this is due to the restrictive choice of d and a smaller d would bring the number of days for the jet-groups closer to the rest. I would suggest the authors, as already previously mentioned, give some further support for this choice. If a smaller d than 0.5 is problematic, is this really due to the transition periods as the text of the authors seems to indicate (line 85)? If this is the case, maybe it is worth mentioning more specifically what the problem about the transition period is (are those so frequently that they can make up nearly 50 percent of the time?). If d= 0.5 was also chosen in the original paper and discussed in depth, I would suggest to explicitly cite it in relation to this aspect.

We will add the information about correlation from the Figure caption into the plot.
As stated before, we will test the sensitivity of the choice of d. With a smaller d more days will be classified, while by applying no restriction, all days will be classified.
A clustering approach of continuous fields such as the atmospheric state will lead to transitions that are not abrupt (e.g. Michel 2011 or Franzke 2011). Beside the transitions days, we expect also to have days that do not resemble any of the clusters, and therefore would also belong to the unclassified days. For example, it was shown by Shaller et al 2018 (using weather regimes) that ~26% of the heatwave days in the European-Russian region occurs within undefined regimes (their Figure S1).

The number of unclassified days per season (Table 1) are 34.6, 51.9 and 41.4 for the NAO, blocking, and jet, respectively. There are more unblocked days than undefined jet, but the jet days are divided into 5 categories instead of 3.

Franzke, C., Woollings, T., & Martius, O. (2011). Persistent circulation regimes and preferred regime transitions in the North Atlantic. *Journal of the atmospheric sciences*, *68*(12), 2809-2825.

Michel, C., & Rivière, G. (2011). The link between Rossby wave breaking and weather regime transitions. *Journal of the Atmospheric Sciences*, *68*(8), 1730-1748.

Schaller, N., Sillmann, J., Anstey, J., Fischer, E. M., Grams, C. M., & Russo, S. (2018). Influence of blocking on Northern European and Western Russian heatwaves in large climate model ensembles. Environmental Research Letters, 13(5), 054015.

lines 200-201:I see, very nice. Maybe it is worth already mentioning this while introducing the reconstruction method. However, this is for the composite of the full climatology. From what I

mentioned before, my assumption was that for individual seasons with dominance of specific classifications (like N-jet), the anomaly of the unclassified cases will show a similar pattern. To estimate this effect a comparison of the variance would be helpful to get a deeper insight. Having a very small value of the anomalies for the climatological composite is nice to conclude, that the remaining unclassified cases indeed represents climatology, but it would also be nice to have a similar short comment about the variance. If it is small then the reader knows that there is absolutely nothing to worry about the procedure in this manuscript. If it is not really small, the authors could give a short explanation why they think that is (no optimal way of defining the classifications because of some cutoff problem between getting the relevant patterns and reducing the variance of the unclassified cases?).

We will add the plot of the NAO neutral, unblocked and undefined day in the manuscript/ supplement. We do not expect that the variance is necessarily small (see lines 249-254), as for example we expect a lot of variability in the precipitation signal from two cyclones that occurs in the same clusters and follow a similar path, due for example to differences in their strength, frontal structure, intensification rate, moisture uptake.

lines 235-236, Figure S2: Oh, it seems I did not understand the uncertainty estimation correctly. I thought that this resampling of the 35 years is done to either observation or reconstruction to get an estimate of how significant the resulting CE-value is. But in this case, I would assume that for the areas of very high correlations, some of them would also show very different values (lower) for the bootstrap mean, because the reconstructed time series was already capturing well the observations, whereas a resampling of the years will not do so. SoI'm slightly puzzled about the near identity between CE and bootstrap mean, suggesting that I don't understand what was actually done. Maybe the authors can therefore explain in a bit more detail how the uncertainty estimation does or what it is for?

We always use the same pair of observation and reconstruction but randomly sample the pairs to build two time series of 35 years. For example, a sample of the time series could include the same year several times and miss other years. So the bootstrap tells us how robust is the calculation of the CE compared to the reference period (i.e. the 35 years of ERA-Interim).

line 262:That wasn't clear to me. I thought the scaling factor was always used after being introduced in the method section. Not sure I have missed this while reading through the text, but I would suggest to make this clear while showing CE for the first time (if not done and I just missed that).

We decided to introduce the scaling factor only in section 3.3. In doing so, the reader will only be confronted with the scaling factor once we use it, and this hopefully will clarify this point.

lines 274-275, Fig. 10:I would suggest to include the 0-line in the anomaly plots, which makes the reading of those plots much easier. This would also highlight the issue of the blocking classifications, that this one does for some regions seems to be only able to capture negative anomalies (10b, d and h). Further, it seems a bit confusing when talking about amplitudes while interpreting this plot. In my mind I quickly switched to my usual interpretation of amplitude, representing the strength of the anomaly. This was confusing, because e.g.Fig. 10b and d seem to indicate that for the blocking reconstruction negative amplitudes of the anomalies are being overestimated while not being able to represent positive anomalies. But here amplitude refers to a from the CE calculation, right? I would suggest to mention this again explicitly, which makes following the interpretations easier.

We will add the 0 line to Figure 10. We also change the formulation in the text. Figure 10 shows the reconstructed magnitude of precipitation and temperature, so the "usual" interpretation. It is not a = std(observed)/ std(predicted). We will be sure to make this point clear.

2.3 Discussion and Conclusion

line 286: Although, the better reconstruction is more focused towards western Europe, whereas temperature is better reconstructed at more eastern and southeastern Europe (Fig. 9). So what does in general mean, a higher mean CE value, occurrence of very high CE values or more points showing CE >0.25? I think the authors could present this a lot more convincingly, if they could attribute a quantification to it, using some sort of simple measure as mentioned above. Just by looking again at Fig. 9 this doesn't seem to be so obvious. I appreciate the authors give some reasoning behind the differences between temperature and precipitation in this paragraph, which represents some explanation beyond just showing the results of this correlation/CE test.

We thank the reviewer for this comment, which raises an interesting point of what constitutes a "better" reconstruction, which will depend on the specific question being asked. As we wish to avoid presupposing the questions people might use such reconstruction methods for, and therefore putting our judgement on what counts as "better", we will change the wording of this section, for example: Precipitation shows substantially higher coefficient of efficiency values over many regions in western Europe (values over 0.5 found in X land grid boxes) than are found for temperature (no values over 0.5 anywhere in the domain for blocking and jet clusters.reconstructions).

line 305: So what is the actual main message of the manuscript. As long as one chooses classifications which are associated with large scale flow anomalies (NAO, blocking, jet), one will be able to capture to some extent the seasonal large scale anomalies over Europe. But this study does not come to a conclusion what metric is best and it also depends on what someone might be interested in (precipitation or temperature). So the main message is that there is no perfect metric to capture both temperature and anomalies? A possible limitation of the reconstruction could be due to extreme events, but those are not considered in this study.

The review is correct, there is no perfect metric to capture both anomalies and the performance depends on the specific region we are interested in. We apologise if this message did not come across that clearly and will add sentences to the conclusions section to emphasise this point.

lines 298-304:I think I made this point already earlier: isn't this (necessity of including a scaling factor) also a problem of the reconstruction method? If one would only use NAO values above 2 or below -2, it would be obvious that the reconstruction would be underestimate the amplitude, wouldn't it? So going down with this threshold to 1 would probably do a better job in reconstructing the seasonal anomalies, because more seasons will be represented within this class and more variability is possible. Whereas defining a class of NAO with values above -5 would do a very bad job, because it will only reproduce climatology. So there is some sort of useful cutoff, or the classification are already defined in a useful way (by EOF, clustering, etc are they already constructed in a way to capture the variability). So I assume my question is, if this shouldn't be rather discussed with more focus on the method / discussion of how necessary a scaling is to start with? Further, I think it was not fully clear in the manuscript which definition of CE was used (I raised this point also in another comment). It wasn't clear to me that the first results were for the non-scaled CE values (sorry if I just missed it). But also the choice of CE (Equ. 4 or Equ. 6) was not fully clear to me. If I am not mistaken, Equ. (4) was taken to calculate the CE values, right? Was Equ. (6) also used somewhere? In general, what is the reason to use Equ. (4)? Since Equ. (6) doesn't really need more calculations or is too complicated, it is not obvious for me why exactly the simplification was used? Is the reason behind this the identification of a useful scaling (Fig. 8)? I think the authors could make this a bit clearer. Also, I hope I didn't miss it, but did the authors compare the two calculations, how different are the results? This information could be useful to justify the general use and handling of Equ. (4).

This is a very interesting comment. Actually, we do not think that it is obvious that using "NAO values above 2 or below -2" instead of 1 "would underestimate the amplitude" because both the frequency of occurrence but also the strength of the anomaly patterns are expected to change.

We will do further analysis to attempt to better understand the need for the scaling factor, including sensitivity tests to the threshold. For the NAO we have already repeated the temperature reconstruction analysis using a regression approach, allowing us to explore some of these questions. This included calculating the NAO related temperature anomalies by regressing daily temperature anomalies onto the daily NAO time series using all days, using positive and negative NAO days separately (i.e. calculating 2 separate regression patterns), and using positive and negative NAO days separately but excluding days with an NAO index magnitude of less than 0.5 standard deviations, i.e. excluding weak NAO days. These approaches showed similar regression patterns, reconstruction correlations, and all require a scaling factor of around 2 to maximise the coefficient of efficiency. We will continue this analysis and add relevant points to the revised manuscript to better explain the need for the scaling factor and discuss which reconstruction methods may require such a factor.

For the definition of CE: we have not used the simplified version to calculate the CE, but the full equation (so equation 3, which is equivalent to equation 6, just expanded and rewritten). Equation 4 is used in Figure 8 (white lines) and for calculating the scaling factor. We apologize if this point was not clear and will make sure to clarify this in the manuscript.

lines 305-314 and lines 315-324: Really interesting aspect in those two ara graphs. But the comparison between NAO (with large region) and jet (with smaller region) is difficult, because of different underlying variables. Further, a larger region does not necessarily help to get better correlations, as large part of the NAO region is also far way from Europe. Therefore, the argument of the NAO region has the advantage to capture also the European region is not really convincing, as the NAO, as mentioned by the authors, also only includes 2 classifications for the whole region. Apart from this, the jet-region also includes part of western Europe, and as Europe is part of the west wind regime, it is not really surprising that the upstream information has an impact on the immediate downstream weather response. Extending the jet-region eastward would be very interesting to see, if this indeed does substantially increase the reconstruction skill, but I would assume this is not clear. So this point is very interesting, but it was not investigated here. Same is true for the next interesting point (number of classifications for one variable), also very interesting but also not included in this manuscript. The question about the relevant or optimal number of classes for the reconstruction exercise does sound also really interesting. But it is not clear to me, what the example of the 30 MSL classifications tells the reader, as at some point maybe too many classifications could be reducing the skill in a similar way as extreme events could be problematic. The more classifications, the more relevant individual extremes could become, because their impact will be felt more strongly during individual seasons.

The main purpose of the study was to compare the ability of three classification methods (NAO, blocking, jet configurations) to reconstruct seasonal climate anomalies. While it is for sure interesting to investigate how reconstruction can be optimized (e.g. following the suggestion mentioned by the reviewer), such analyses are beyond the scope of this study. However, we can certainly address some of these issues in a revised manuscript.

With the example of 30 MSL, we wanted to discuss that retaining too much information (i.e. using 30 MSL and all days, which also includes quite some noise) does not necessarily lead to better skill than using only 2, 3 or 4 patterns. Therefore, too much information does not necessarily translate into better reconstruction skill. We can clarify the purpose of this example in the manuscript.

To illustrate why we feel the region selected for analysis could play a role, here is an example. The NAO region covers much more of Europe than the jet domain. Since the NAO is calculated by EOF analysis that reveals the pattern that maximises the variance in that region, it can be expected that using a region that extends eastwards of 0° will better capture the variability over Europe and Scandinavia. To illustrate this, we calculate the NAO index using daily SLP data for the NAO domain (1st plot) and the jet domain (2nd plot). The NAO pattern is visible from both analyses, but the patterns (EOFs) are not identical and we see that the influence of the NAO extends eastern of 0° (left plot). When looking at the temporal evolution of PC1, we see that even if the two time series are correlated, the days that will be classified as NAO+/- (exceeding the 0.5 standard deviation, 3rd plot) will not be exactly the same and could lead to different reconstructions.

[Figure]

lines 325-330:Same as for lines 305-314 and 315-324. The authors nicely present a way of how their results can be interpreted, where relevant impacts of the method on the results might occur or how specific changes in the classification setup could have an impact. I find this discussion very helpful, but I am somehow missing the part where the authors are highlighting the results of their study. Main issue with this point can be found in my general comment.

We will address this point while addressing the general comments.

---

## Author Comment (AC2)

**Revision of wcd-2021-6 "Reconstructing winter climate anomalies in the Euro-Atlantic sector using circulation patterns" by Madonna et al.**

We would like to thank the reviewers for their constructive comments and suggestions.
In response to the main issues raised by the reviewer, we plan to:
- provide a sensitivity analysis for the choice of the thresholds (e.g. d)
- clarify how the CE has been calculated, discuss the assumptions and their validity for our calculation, and improve the presentation and application of the scaling factor.
- better highlight the importance of the study and its implications

Reviewers' comments are in black, our replies are in blue.

**Reviewer 2:**

The paper presents an analysis on the ability of different circulation patterns frameworks to describe the seasonal wintertime precipitation and temperature anomalies over Europe, using ERAInterim reanalysis data. Three different frameworks are used: the two NAO phases, blocking over three different regions and five Atlantic jet stream clusters.

The paper is clear and well written in all parts, and covers a current hole in the literature regarding circulation patterns, presenting original results which can be very useful for the community. I therefore recommend the article for publication in Weather and Climate Dynamics, once the following minor comments are considered.

General comments/suggestions

- I was very pleased to see the topic of this work, which I think was missing in the literature, and really enjoyed its reading. As a minor comment, I don't really understand the authors' choice not to consider - amongst the sets of circulation patterns - the "classical" four Euro-Atlantic weather regimes framework, which has been (and still is, to my knowledge) the most used in literature. I think it would be in the authors' interest to show the main results (e.g. Fig. 6 and 7/9) also for the 4 regimes framework, which would be of interest for many works on regimes when discussing the related impacts. I respect the authors' choice and I'm not asking to repeat the analysis for the k=4 case, but maybe some comments in the text and conclusions referring to the correspondence between the k=5 and k=4 regimes would help the reader "translating" the results to that framework.

We agree that the four Euro-Atlantic winter reglmes widely used in the literature. We compared the different k in a previous study (Madonna et al 2017), using the jet latitude index (3 regimes), the classical weather regimes and the jet cluster using k=3,4,5. Even if using different variables (zonal wind vs. geopotential height) and domains, in Madonna et al 2017 we showed the 4 jet clusters correspond well to the 4 weather regimes. Using 5 jet clusters, the analysis revealed that the zonal regime is a combination of central (48%) and tilted (37%) jet (their Figure 10). We are happy to include and comment and "translate" our results to this perspective.

- As discussed by the authors, one of the problems of the reconstruction is the underestimation of the seasonal anomalies, due to the limits of the mean composites. Maybe a simple estimate of how much does the variability impact the reconstruction in the different regions would be given by the ratio between the standard deviation and the mean of the composite patterns. Just a suggestion.

We thank the reviewer for the suggestion. We can compute the ratio std(x)/mean(x) for the composite pattern. We expect to have large values of standard deviation within each pattern, as noted in lines 249-25, and this can be a reason for the underestimation of the amplitude of the reconstruction.

Specific comments

- L35. "..the North Atlantic jet stream can assume five different configurations..": the number of clusters to be considered is a matter of debate in literature and probably will not lead to a conclusive "best number", since all choices retain some level of arbitrariness. Please acknowledge here that this is the authors' choice, while other choices for the number of regimes are possible (e.g. as in Madonna, 2017).

We agree with the reviewer and will add a comment to acknowledge that.

- L70. Please briefly comment on the choice of 10%. Have you tried with other thresholds? Are the results sensitive to this choice?

The choice of 10% (i.e. 0.1) is to a certain extent arbitrary and arose as we did not want to select days where only a few gridpoints were blocked. The Figure on the left shows a histogram with the fraction of gridpoints (x-axis) in the "Greenland box" for all days with blocking, while the Figure on the right shows the fraction of blocked gridpoints for a subset of 120 days. The 10% (i.e. 0.1) threshold is marked by the red line. From the time series (right), we see that most of the points with values below 0.1 are followed by points with a larger threshold, which means that the block is building and the 0.1 threshold captures it only at a later stage. Translated to the occurence of blocked days per season, we expect that the absolute frequency depends on the threshold, but not the number of blocking events.

[Figure]

[Figure]

- L83. How is *d* calculated? Please show the formula here or add a reference.

d is the normalized inverse Euclidean distance from the centroid. The Euclidean Distance (E) for a point x from the centroid in a n-dimensional space is defined as $E = \sqrt{\sum_{i=1}^{n}(centroid_i - x_i)^2}$ .
d is then the inverse of the normalised distance, d=1/Enorm, where the normalisation is Enorm = E /max(E).

- L86. Is this method and the threshold of 0.5 used elsewhere? If so, please add a reference here. Instead please briefly discuss the choice and its possible impacts on the results.

The choice of d=0.5 is somehow arbitrary but was introduced to be sure to capture only days that belong to the given centroid. We will provide some sensitivity tests to discuss the possible impact of this choice.

- L105. How close to a zero anomaly are the undefined days? I imagine the mean is very close to zero, but this may also come from cancellation of opposite anomalies. I'd suggest to show the standard deviation of the anomalies for the unclassified days (in the supplementary), to assess whether their exclusion might impact the skill of the reconstruction. The underestimation of the amplitude of the seasonal anomalies might also be linked to the filtering, since the denominator in equation 2 is always 90, but the number of assigned days is usually much less.

The anomalies of the undefined days are close to climatology. We will add them together with the standard deviation for the composites of all categories in the manuscript or supplement. The standard deviations in some regions are as large as the mean values. This can be understood from a synoptic perspective: for example, if two storms follow a similar path, the exact location where the most intense precipitation falls can be quite different.

From preliminary sensitivity tests, we don't believe our final results will change much with the number of defined/undefined days. However, we will do a more thorough sensitivity analysis to better understand the underestimation of the scaling factor in our revision.

- L129. CE>0.25. Is this threshold used elsewhere? If so, please add a reference.

A reconstruction that had no bias in the variance (a =1) and CE = 0.25 would have r=0.63, or explain 39% of the total variance; given 35 years of data (degrees of freedom), this correlation would be significant at p=0.0001. For a = 0.5, r=0.50 (p=0.0008); for a=0.25, r=0.63 (p=0.0001). Alternately, a reconstruction with r^2 = 0.5 and a=0.75 would have a CE = 0.57. As mentioned before, we will provide some more explanation for how to interpret the CE in the methods, along with some references.

- L135. Is the "/" a typo?

Thanks.

- Section 2.2.2. I found it quite hard the reading of the section before reading Section 3.3 and looking at Figure 8. To make this easier, I suggest to explicitly refer to Section 3.3 and Figure 8 in the text. Also, a possibility would be to move Section 2.2.2 to the beginning of Section 3.3, since the scaling factors are not used till then.

We agree that the introduction of the scaling factor in the method, but their application only in 3.3. make the reading and understanding of this part hard. We think that moving its introduction at the beginning of section 3.3, as suggested by the reviewer, would improve the clarity of the manuscript.

- L138. "small". with respect to the mean?

with respect to standard normal distribution

- L163. "are placed to allow easy comparison": has some automatic matching of the wind patterns been performed (e.g. maximizing the pattern correlation)?

In Madonna et al. 2017 the wind classification has been explicitly compared with blocking and NAO. There has been a temporal match (i.e. by looking at the number of overlapping days in each definition). A spatial match has been performed only between the different jet clusters in Madonna et al 2017 (i.e. for k=3,4 and 5, their Table 1).

- L174. The S-jet cluster is made up by less days, so that would probably enlarge the anomalies.

Let's assume that the S-jet is a subsample of the NAO-. If the S-jet days are equally distributed within the NAO- days, the anomalies for S-jet should not be enlarged just because we use less days. But the anomalies would be enlarged if the subsample of S-jet belongs to the upper tail of the NAO- distribution, or reduced if they are located in the lower tail. Therefore, we do not expect the anomalies to be stronger just because fewer days are used.

- L178-179, L197. In the comparison with NAO+, the usual 4 regimes would look more natural to me. Probably a sum of the tilted and central jet states would show similar anomalies, as it is said in the text, and also correlate better with the NAO+ timeseries. It might be worth adding a comment on this in the text.

We will add a comment in the text and also add a discussion to Madonna et al. 2017, where the 4 classical weather regimes and the jet clusters (k=4 and 5) have been compared. In fact, the NAO+ regime (i.e. the zonal regime) is a mix of central (48%) and tilted (37%) jet (their Figure 10).

- L200. The unblocked days are different from the undefined/neutral days, so I won't put them in the same category. I expect (at least part of) them to correlate with positive NAO and have similar anomalies. I suggest to add a comment on this in the text.

We agree that the unblocked day category includes days that belong to the "zonal flow/NAO+ regime" and days that are undefined. This choice was intentional, as the goal was to assess how much blocking episodes influence the seasonal signal. We will add a comment on that in the manuscript.

- L201. See comment at line 105. I think it would be interesting to show the mean (even if close to zero) and standard deviation for these days in the additional material.

We will add these Figures in the supplement.

- Section 3.2. Have the reconstructions been performed for the wind also? Table 2 and L260 seem to suggest so. Why are the results not shown? It might be worth commenting on this briefly in the text.

Yes, we did that for wind as well. We presented only precipitation and temperature on the one hand because as seasonal means they have a stronger surface impact than the wind, and on the other hand because the winds/circulation is somehow included in the definition of the categories. We are happy to add the CE for wind in the supplement and add a brief comment in the text.

- L216. "that" -> than?

Thank you

- L224. Also over France the blocking method has substantially better skill.

The sentence has been modified to cover Spain and France.

- L227-8. "...regions with poor correlation skill mostly have low temperature variability..". Is this referred to the NAO phases or to all methods? I don't think this is true in general. For example Spain and southern Italy show low temperature variability but good skill for blocking/jet regimes.

The reviewer is right, and we will reformulate this part.

- L240. A large skill is also apparent over North Africa/the Mediterranean, I'd add a comment on this in the text. Also, if the figures are already available, it might be worth showing the equivalent of Figure S3 for the correlation also.

We will add the correlation for a larger domain also in the supplement and a comment regarding North Africa in the text.

- L292-297. I'd comment on the case of France as well, which is well represented only by the blocking framework.

We are happy to include a comment on that.

- L302-304. Also, this might be related to the filtering, see comment at line 105.

We will do some additional investigation on the scaling factor, as those suggested by the reviewer.

- L311. The "classical" 4 Euro-Atlantic weather regimes are usually calculated using larger domains, extending up to 40 degrees east. Do you think this could increase the skill in central/eastern Europe?

If much of the variability over easter Europe is captured by the first 5 EOFs, that could be the case. We know that the circulation over the North Atlantic plays an important role for the European region and that there is a good correspondence between the jets (k=4) and the weather regimes (from Madonna et al 2017). Using a metric that covers a larger domain might capture more of the variance over eastern Europe and increase the reconstruction skill. However, we have not tested that yet, but could be a nice additional analysis.

---

## Author Response (AR1)

**Revision of wcd-2021-6 "Reconstructing winter climate anomalies in the Euro-Atlantic sector using circulation patterns" by Madonna et al.**

We would like to thank the reviewers for their constructive comments and suggestions.
In response to the main issues raised by the reviewers, we made the following main changes:
- We provide a sensitivity analysis for the choice of the thresholds (e.g. d) in the Supplement
- We clarify how the CE has been calculated, discuss the assumptions and their validity for our calculation, and improve the presentation and application of the scaling factor. This led to some changes in the results, in particular for the blocking classification.
- We reorganize the discussion and conclusion, separating the two parts. We now include a discussion about the need for the scaling factor. The conclusion is more concise to better highlight the importance of the study and its implications.

Reviewers' comments are in black, our replies are in blue.

**Reviewer 1**

**1 General comments**

The authors present a really nice study about the reconstruction of winter climate anomalies in the Euro-Atlantic sector using different measures to classify large scale flow anomalies. This is done with the broadly used global pattern index of the NAO,widely recognised as an important factor to influence winter anomalies over Europe.The authors further use blocking regions and jet classificationsin the North Atlanticto investigate how well European winter anomalies, in temperature and precipitation,can be reconstructed by those groups (NAO, blocking jet). The author do not do this solely by using simple correlations, but also another measure also accounting for the represented temporal variability (amplitude) of those groups. The manuscript is very well structured, everything is quite well explained and results are presented in a very coherent way, guiding the reader through the individual steps of the results and how they are connected. The authors also spent enough time to explain the methods and give the reader the necessary information (e.g. including Fig. 1) to understand the applied diagnostics without having to check the references for the essential informations. All in all, very nicely done.I only have two major issues.

The smaller issue concerns the application of the diagnostics in the result section. It wasn't always fully clear which of the specific representation of their diagnostics is applied, e.g. concerning the CE representation. While reading the result section I thought the authors always use the scaled reconstruction (with β), only after they explicitly mentioned that they introduce now the scaling did it become clear to me that they didn't use it before. Also, the authors could have clarified how Equ. (4) and Equ. (6) are used in this manuscript. If I'm not mistaken, only the simplification (Equ. 4) is used. And although highlighting the different Equations, the authors did not make a statement on the impact of the different choices, except for saying that the assumption of a zero climatological mean of the reconstructed anomalies should be fullfilled. More specific comments about this can be found in the "Specific comments" section.

We agree that in section 2.2.2, the description of how the diagnostic is used was not totally clear. Linked to the comment of reviewer 2, we moved the description of the scaling factor to section 3.3, where we use it. We tried to better explain which equations we used and for which parts of the analysis throughout. To compute the CE we now use equation 3 and not the approximation, which

makes no difference for the NAO and jet clusters, but changes the result for blocking. Modifications have been applied also to section 2.2.1 to improve clarity.

My main issue with the manuscript is that the authors maybe did not the best job in highlighting the importance of their study and the usefulness of their presented results. The final discussion is full of very interesting aspects which are closely linked to this study, but which weren't analysed. This includes for example the extension of the regions to identify the classifications of a specific group (e.g. the limited eastward extension of the jet-region), the number of classifications within a group, the impact of extreme events for this kind of climate reconstruction. Also, the main conclusion I did get from this is that somehow all those groups are able to some extent to reconstruct climate anomalies, but it is also not clear which one is best. The thing is that it cannot be surprising that any kind of large scale feature which represents a large part of the variability of winter anomalies (variability in the same variable) is very likely able to identify large scale seasonal anomalies in other variables like temperature or precipitation, as those variables are linked to each other (no rain within a blocking high, storms moving along the jet, etc.). I think the authors could made clearer what there main message is, why it might be important to use different classes. How can those results interpreted and maybe applied to other fields (e.g. if climate models are able to capture the general jet behaviour over the Atlantic this might be not enough as the exact representation of the established blocking periods decides on whether or not the model can have skill for specific parts of Europe in reconstructing the climate). Not sure the above examples make any sense at all, but I try to say with that that the authors could put some more effort into selling their results, why is this study so useful (I think this manuscript is slightly missing this). This is probably even more important because the authors restricted their analysis strongly to the choices they have done. As previously mentioned, and also raised by the authors, there are so many interesting modifications of the setup of the study and modifications of the choices which weren't done by the authors, although probably having important impact on the results. Also are there a few questions arising from the conclusions of the authors, e.g. can the authors quantify the relevance of remote regions (Pacific jet configurations, ENSO, North American or Pacific blocking) for the European winter climate reconstruction (as authors highlight the importance of jet classification region to European climate response). To summarize, I think if the authors could highlight more the important aspects of their results beyond the connection that large scale flow anomalies are connected with large scale climate response, the impact and relevance of this study could be improved. One example for highlighting the results of this manuscript could be the connection between the different groups of large scale flow representation. Not sure how new and innovative this is, but I really appreciated how well the authors presented the connection between those large scale flow anomalies (link between NAO, blocking and jet). This was really nicely presented (e.g. Fig 2-4 and Fig. 5). Not sure if this was already done in a similar fashion in other studies, but I expected this to be highlighted at the end, because it nicely show the connections and links between those different groups (not only using only one group, like NAO, but compare it with different large scale flow realisations and show how those are connected). Maybe one could have also interpreted how this can be used to understand the similar usefulness to reconstruct seasonal anomalies or how specific aspects/regions cannot be as well reconstructed as others for a specific group.

We thank the reviewer for the thoughtful comments and suggestions. We reorganized the discussion and conclusion section. Now the conclusion is more coincese to better highlight the importance of the study and its implications. Building from the reviewer's line of thought, the study could help to understand, in climate change scenarios, what types of weather events contribute to changes in the climatological temperature and precipitation (l. 389-393 and 20-23).
We also expanded our discussion on the sensitivity of the results to some of the analysis choices we made, as suggested by the reviewer, such as the threshold d (in the supplement, section A) and the regression approach (in the discussion, l.343-363).

2 Specific comments

2.1 Introduction

lines 29-30: I wonder if "obstructs" is the appropriate word, as specific locations of a blocking high can also lead to a northward or southward shift of the jet - so changing the path of the jet but not really fully blocking it. So maybe it would be better to specify "obstruction"in terms of obstructing the climatological position of the westerly flow (which of course will have an impact on seasonal anomalies in temperature, etc.

We reformulated the sentence to acknowledge the possibility of a flow deviation. "In contrast, blocking is the presence of a persistent and stationary high-pressure system that obstructs or deviates the westerly flow."

line 36: Not sure how well the term storminess is established. I think it would need a short explanation. Therefore I would suggest to remove "with distinct patterns of storminess", because the sentence works also without this term.

We replaced "storminess" with "patterns of storm propagation,".

Table 1: Maybe replace "NAO values above (below) 0.5" with "NAO values above 0.5 (below-0.5)".

Thank you for the suggestion, which is now implemented.

lines 83-87: Reading this, one might wonder about the choice of the distance d= 0.5 if 46 % cannot be classified. This does sound like only the days very close to a centroid are chosen, which seems fine to create clear jet classifications. However, the explanation is somewhat surprising, namely that the large group of undefined cases is in particular occuring during the transition days, which would mean that either the different jet classification are associated with frequent changes or that the transition phase between the jet classifications is rather long. Is the choice of d=0.5 the same as in Madonna et al(2017)?

In Madonna et al 2017 all days are used, as the purpose of the study was to reconcile the weather regime and jet variability frameworks. It is not exactly that 46% of the days can not be classified, but that 46% of the days share similarities with more than one cluster. This often happens during transition days, as illustrated in the table below. Previous studies also showed that transitions from a jet state to another are not instantaneous (e.g. Franzke et al 2011) and occur relatively frequently (~ median persistence of a jet regime is ~ 1 week, e.g. Franzke et al 2011, Madonna 2017). However, not all the unclassified days are transition days. For example, extreme events that do not resemble any of the clusters can belong to the unclassified category.

As an example, we show the inverse distance (1/d) from the centroid for 1981.01.14 to 1981.01.21:

| 1/d | N | M | T | C | S |
|------------|------|------|------|------|------|
| 1981.01.14 | 0.79 | 0.04 | 0.05 | 0.08 | 0.03 |
| 1981.01.15 | 0.61 | 0.06 | 0.09 | 0.17 | 0.06 |
| 1981.01.16 | 0.49 | 0.07 | 0.12 | 0.23 | 0.09 |
| 1981.01.17 | 0.43 | 0.07 | 0.15 | 0.23 | 0.12 |

| | | | | |
|---|---|---|---|---|
| 1981.01.18 | 0.42 | 0.07 | 0.19 | 0.19 | 0.13 |
| 1981.01.19 | 0.44 | 0.08 | 0.22 | 0.14 | 0.13 |
| 1981.01.20 | 0.46 | 0.08 | 0.26 | 0.09 | 0.11 |
| 1981.01.21 | 0.47 | 0.08 | 0.28 | 0.07 | 0.1 |
| 1981.01.22 | 0.46 | 0.09 | 0.31 | 0.05 | 0.09 |
| 1981.01.23 | 0.4 | 0.1 | 0.37 | 0.04 | 0.08 |
| 1981.01.24 | 0.26 | 0.13 | 0.5 | 0.04 | 0.08 |
| 1981.01.25 | 0.12 | 0.15 | 0.63 | 0.03 | 0.07 |
| 1981.01.26 | 0.07 | 0.19 | 0.65 | 0.03 | 0.07 |

This period includes a transition from a northern (N) jet to a tilted (T) jet. Some of the transition days are quite similar to more than one centroid, while others have a d < 0.5 and are counted as undefined days.

We agree that the choice of d=0.5 is rather restrictive but was introduced to be sure to capture only days that clearly belong to a certain cluster. Linked to the major comment of reviewer 1 and also a comment from reviewer 2, we performed a sensitivity analysis to the choice of d, which is presented in the supplement.

References: Franzke, C., Woollings, T., & Martius, O. (2011). Persistent circulation regimes and preferred regime transitions in the North Atlantic. *Journal of the atmospheric sciences*, *68*(12), 2809-2825.

reference also to section 2.2.2:
Also having read the part of Arec, this jet classification seems relevant, as stricter classification choices (larger d) leads to less days recognized as jet classification. Since Arec is normalized by the full number of days (90), this could lead to an increase in the amplitude error (for CE, Equ. 4). So introducing the beta factor (Equ. 7) is essential to account for this and therefore corrects possible "missing" members of specific classifications, or am I mistaken? But in lines 137-139 the authors only mention the problem of a non-gaussian distribution which "requires" a correction with this β-factor. Not sure I made myself clear, therefore I include a short example. If I could assume that the seasonal anomalies are mainly driven by a N-jet and a S-jet, the choice of a larger d can lead to a underrepresentation of the anomalies for seasons mainly associated with either a N-jet or S-jet, meaning that the associated a-value will become smaller and with it will affect CE. So the β-factor will boost the dominant anomaly. If this is true, maybe this should also be included as a reason in lines 137-139.

We thank the reviewer for the comment. We tested the sensitivity of the anomalies to the choice of d and how this influences the β-factor. These results are shown in the supplement. Even if all days are used for the reconstruction, we still need a scaling factor β. We also perform an additional analysis with a regression approach, focussing on the NAO and temperature, to try to understand why we need the scaling factor beta (new paragraph in the discussion section, I. 345-363):

"Using a simple ordinary linear regression on daily NAO and temperature values we find relationships (◦ C per unit anomalous NAO index) very similar to those found by the composite method shown in Fig. 4. A reconstruction is then made by multiplying the regression pattern by the mean NAO value for each season. In this regression approach, all days are used in the reconstruction (compared to about half the days in the composite approach). However, the correlation and CE values of the two reconstructions are very similar, and both require a scaling factor to maximise the skill. This suggests that the need for a scaling factor is not linked to the omission of information (i.e. the neutral NAO or undefined days).

If, instead, the regression between daily temperature anomalies and daily NAO values is calculated using a weighted orthogonal distance regression (using the python package scipy.odr), the regression relationship changes - the slope of the linear fit generally increases. An example is shown in Figure 11 for Bergen (cf. black and blue regression lines). This increase in regression slope means that the reconstruction amplitudes increase, and there is less need for a scaling factor. The weighted orthogonal distance regression takes into account "noise" in both the temperature and the NAO values, while ordinary least squares considers the values of the independent variable (in this case the NAO) to be exact (e.g. Wu and Yu, 2018). This noise may be related to a lag/lead relationship between circulation patterns and surface weather anomalies and/or uncertainty in the connections between the NAO circulation anomalies and surface temperature. In Fig. 11 the mean composite values for this grid-box are shown in the cyan shapes, falling on the ordinary least squares regression line. Values for using the median of the composites are also shown in purple, demonstrating that using the composite median would not remove the need for the scaling factor. This suggests that the need for the scaling factor in the composite and ordinary least squares regression reflects the large variability in temperature and precipitation within each classification pattern (cf. Fig. S7 and S8)."

lines 95-97:Why do the authors not apply the same anomaly calculation to all variables? The position of the jet does also follow a seasonal cycle, following the location of the temperature gradients. Therefore, what is the reason for not applying the same anomaly calculation? If there is a reason for doing so, e.g. signal otherwise too noisy because of the mentioned strong fluctuations, I think this is worth mentioning. Otherwise the reader will wonder about the different procedures for the different variables.

We agree with the reviewer that we could have removed the daily climatology of all variables. The choice to leave in the climatology was mainly because previous studies have shown that the seasonal cycle of wind (see Woollings et al 2014, Figure 3a) is small within the DJF winter considered here. Also, most studies looking at seasonality in precipitation aggregate DJF (e.g. Zveryaev 2004 or Rauthe et al 2013). We mentioned this in the manuscript (l 111-114).

Woollings, T., Czuchnicki, C., & Franzke, C. (2014). Twentieth century North Atlantic jet variability. *Quarterly Journal of the Royal Meteorological Society*, *140*(680), 783-791.

Zveryaev, I. I. (2004), Seasonality in precipitation variability over Europe, *J. Geophys. Res.*, 109, D05103, doi:10.1029/2003JD003668.

Rauthe, M., Steiner, H., Riediger, U., Mazurkiewicz, A., & Gratzki, A. (2013). A Central European precipitation climatology–Part I: Generation and validation of a high-resolution gridded daily data set (HYRAS). *Meteorol. Z*, *22*(3), 235-256.

Equation (3): This refers to Equation (3) and the whole section 2.2.1. According to Equ. (3), CE represents a 2d field and the given equations and text (CE values of 0, 1, etc) are for timeseries at one gridpoint, correct? While first scanning the manuscript, I expected CE tobe a single-value metric

to capture the overall skill of the 2d reconstruction (because the examples were for one value, e.g.CE= 1), but in section 2.2.1 references are only made to time series, so CE must have values at each longitude and latitude. To make this fully clear and prevent confusion, CE could be written in the same way as Arec, as CE($\varphi$,$\lambda$). Alternatively a short sentence mentioning this would be helpful.

The reviewer is right, CE is calculated at each grid point. We added this information in line 136.

Section 2.2 Seasonal reconstructions: The underlying assumption of this procedure is that the unclassified days more or less represent the climatological mean, otherwise the large undefined classifications (with representing partly more than 50 %) would account for large anomalies that this procedure will never be able to capture. This means for normally distributed quantities, as long as there is no mixing of different classifications (e.g. one jet classification and one blocking classification), that this method tests how well the undefined classifications represent the climatology (i.e. is there a missing classification with strongly impacts seasonal anomalies), right? Maybe this is worth mentioning (if true), because this could get lost in the text with all the description of weighting procedure and combination of classifications? However, non-normally distributed quantities would also lower the reconstruction skill. So probably these two aspects can lead to a reduced reconstruction skill, non-climatological rest-classifications and non-normal distributions. Maybe worth mentioning here, because both aspects are a necessary assumption for this method.

This is correct: the anomalies of the undefined days are almost zero, which means that the undefined category represents climatology. We briefly mention this later in the text (former l.201). We now included the composited with the anomalies of the undefined days in the supplement (Figure S1) and added a comment (l. 128-132).

line 120:Same as in last comment, the assumption hereby is that the undefined classifications are close to climatology, otherwise one cannot assume that the average of p is zero. As this is now part of the definition of CE (Equation 4), I think this should be clarified, or alternatively the step from Equ. (3) to Equ. (4) should be justified.

The reviewer is right, the approximation is valid only if the average of the reconstructed anomalies $(\bar{p})$is zero. This is the case for the NAO and the jet (see Figure S2 in the supplement), but not for the blocking. The CE is now calculated using equation 3 and the approximation is used only for estimating the scaling factor for the NAO and jet. We have clarified this in the manuscript (l.150-151). Also please note that the equation numbers have changed (former eq. 4 is now 5).

line 129:Where does the CE= 0.25 comes from? Is this a usual value indicating good reconstructions, also used in other studies? Or alternatively the authors could give some further insight into the simple function CE. The global maximum of this CE function, as function of a, is at the location r=a, which means that the maximum value of CE for any a is r**2. This would mean that independent of the amplitude error (or a), the choice of CE= 0.25 requires a minimum correlation of 0.5, whereas for a perfect correlation the values of a must be between 0.13 and 1.87 (if I am not mistaken). I would find such a simple consideration of CE very helpful, otherwise I would feel a bit lost when reading about the choice of 0.25 (because initially, I had no feeling what this value means in terms of correlation or amplitude error). What I find slightly confusing about this CE-value is that for exceeding the 0.25 value at low correlations (slightly above 0.5), it must include an amplitude error (a <1) in order to exceed the threshold of 0.25, so low correlations can be corrected by a reduced variance in the reconstructed fields. But the overall conclusion from this function is, that it is a well established, easy measure to evaluate the reconstruction of an original field?

We thank the reviewer for this comment. We complemented the example of the interpretation of CE with the very nice example illustrated by the reviewer. This will definitely help provide readers with

more intuition about what the CE and our definition of a good reconstruction means. We have marked with white dots regions with low correlations (r<0.5) in Figures 7 and 9 and also used only the non-dotted regions to calculate the scaling factor (in line 286-288).

The new paragraph (l. 152-161) reads: "When applied to reconstructions and observations, the CE is a measure of skill in reconstruction that is more restrictive than a simple correlation because it penalizes for both phase and amplitude misfits. For a perfect reconstruction, CE=1. For a reconstruction with observed variance (a=1) that is correlated with the observed time series of seasonal anomalies at r=0.5, the result is CE=0. For a reconstruction that is perfectly correlated with observations but with twice the observed amplitude, we arrive at the same result of CE=0. We consider CE>0.25 to indicate a good reconstruction. A reconstruction that has perfect variance (a=1) and CE= 0.25 would explain 39% of the observed variance (r=0.63); given 35 years of data (degrees of freedom), this correlation would be significant at p=0.0001. For a reconstruction with perfect correlation (i.e. r=1), CE>0.25 can be obtained for amplitude values 0.13< a <1.87. As the CE maximizes when r = a (and max(CE)= r 2 ), a CE of 0.25 also implies a minimum correlation of 0.5 independent of the amplitude error. For this reason, only points with correlation greater than 0.5 are considered when optimizing the CE (see Sec. 7)."

lines 139-140: But using the median would lead to a bias in the overall climatological reconstruction of this variable (sum/average over all seasons), wouldn't it? But this would be fine, because this bias is not relevant for the calculation of CE, is this correct? So what I found surprising while first reading this sentence is that there is no condition that the climatological reconstruction of this method does not necessarily need to represent the climatology. Sorry, not sure this even is a question, or if it is obvious and not worth mentioning.

Linked to a previous comment, we now calculate the CE using the full equation (eq. 3 and new 4), which takes into account the climatologies of the observed ($\bar{o}$) and reconstructed ($\bar{p}$) anomalies (new equation 4), thus accounting for possible non-zero $\bar{p}$.

line 145:"...calculate the centre of mass of the CE...". Is there a weighting by the value of CE involved to calculate the centre of mass, or is this purely by distance in correlation-amplitude space with each value represented by a weighting value of 1 (e.g. what is the"mass" here)? I see, in line 149 the points are restricted to correlation values of above 0.5, which probably suggests that the weights are all 1. I would suggest to be clarify this in the text.

Yes, each grid point with r>0.5 has the same weight. We clarified this in the text in line 277.

line 155: Is uncertainty the correct term here? It does not really test the uncertainty of the calculation of CE, but checks if the resulting CE-values are statistical significant, doesn't it?

With bootstrapping, we resampled the 35-year time series to assess the effect of internal variability on the CE. We show the standard deviation of the bootstrapped CE distribution, but we could instead have plotted a confidence interval (e.g., 5th to 95th percentile) as an alternative way to express the uncertainty (Wood 2004). A classical significant test would only test the null hypothesis, i.e. CE≠ 0. Since we added a sensitivity analysis to d, we decided to move this subsection in the Supplement, where the Figure is presented.

Wood, M. (2004). Statistical inference using bootstrap confidence intervals. Significance, 1(4), 180-182.

2.2 Results:
line 190, Fig. 5:Very nice overview of the seasonal connection between different, already coherently grouped, classifications. It would be nice if the NAO- would also have the associated correlation value with the jet classification, as all the other values have. Now with the missing r-value for NAO-, one has

to check always the figure caption to understand for which variable GB has a correlation of 0.64. I would also include the NAO+ correlations in the legend. As the variable without a correlation value is the one with which the correlation is calculated, there is never any confusion. lines 190-197:Coming back again to the choice of the d-value for the jet classifications. I think it would be helpful if the authors explicitly mention their "large" choice of d= 0.5. As for all cluster groups (e.g. NAO-, GB and S-jet - IWB and N-jet - etc) the jet part represents the smallest part (in terms of number of days), it seems like this is due to the restrictive choice of d and a smaller d would bring the number of days for the jet-groups closer to the rest. I would suggest the authors, as already previously mentioned, give some further support for this choice. If a smaller d than 0.5 is problematic, is this really due to the transition periods as the text of the authors seems to indicate (line 85)? If this is the case, maybe it is worth mentioning more specifically what the problem about the transition period is (are those so frequently that they can make up nearly 50 percent of the time?). If d= 0.5 was also chosen in the original paper and discussed in depth, I would suggest to explicitly cite it in relation to this aspect.

We added the information about correlations from the Figure caption into the plot.
We tested the sensitivity of the choice of d and presented the results in the supplement (e.g. Figure A2). With a smaller d more days are classified, while by applying no restriction, all days are classified. A clustering approach of continuous fields such as the atmospheric state will lead to transitions that are not abrupt (e.g. Michel 2011 or Franzke 2011). Besides the transition days, we expect also to have days that do not resemble any of the clusters and therefore would also belong to the unclassified days. For example, it was shown by Shaller et al 2018 (using weather regimes) that ~26% of the heatwave days in the European-Russian region occur within undefined regimes (their Figure S1).

The number of unclassified days per season (Table 1) are 34.6, 51.9, and 41.4 for the NAO, blocking, and jet, respectively. There are more unblocked days than undefined jet, but the jet days are divided into 5 categories instead of 3.

Franzke, C., Woollings, T., & Martius, O. (2011). Persistent circulation regimes and preferred regime transitions in the North Atlantic. *Journal of the atmospheric sciences*, *68*(12), 2809-2825.

Michel, C., & Rivière, G. (2011). The link between Rossby wave breaking and weather regime transitions. *Journal of the Atmospheric Sciences*, *68*(8), 1730-1748.

Schaller, N., Sillmann, J., Anstey, J., Fischer, E. M., Grams, C. M., & Russo, S. (2018). Influence of blocking on Northern European and Western Russian heatwaves in large climate model ensembles. Environmental Research Letters, 13(5), 054015.

lines 200-201:I see, very nice. Maybe it is worth already mentioning this while introducing the reconstruction method. However, this is for the composite of the full climatology. From what I mentioned before, my assumption was that for individual seasons with dominance of specific classifications (like N-jet), the anomaly of the unclassified cases will show a similar pattern. To estimate this effect a comparison of the variance would be helpful to get a deeper insight. Having a very small value of the anomalies for the climatological composite is nice to conclude, that the remaining unclassified cases indeed represents climatology, but it would also be nice to have a similar short comment about the variance. If it is small then the reader knows that there is absolutely nothing to worry about the procedure in this manuscript. If it is not really small, the authors could give a short explanation why they think that is (no optimal way of defining the classifications because of some cutoff problem between getting the relevant patterns and reducing the variance of the unclassified cases?).

We added the plot of the NAO neutral, unblocked and undefined day in the supplement (Fig. S1). We do not expect that the variance is necessarily small (see lines 249-254 of the original manuscript), as

for example there could be large differences in the precipitation signal from two cyclones that occur in the same cluster and follow similar paths, but exhibit differences in their strength, frontal structure, intensification rate, moisture uptake etc. The composite standard deviation has also been added to the supplement (Figs. S7-8).

lines 235-236, Figure S2: Oh, it seems I did not understand the uncertainty estimation correctly. I thought that this resampling of the 35 years is done to either observation or reconstruction to get an estimate of how significant the resulting CE-value is. But in this case, I would assume that for the areas of very high correlations, some of them would also show very different values (lower) for the bootstrap mean, because the reconstructed time series was already capturing well the observations, whereas a resampling of the years will not do so. SoI'm slightly puzzled about the near identity between CE and bootstrap mean, suggesting that I don't understand what was actually done. Maybe the authors can therefore explain in a bit more detail how the uncertainty estimation does or what it is for?

We always use the same pair of observation and reconstruction but randomly sample the pairs to build two time series of 35 years. For example, a sample of the time series could include the same year several times and miss other years. So the bootstrap tells us how robust is the calculation of the CE compared to the reference period (i.e. the 35 years of ERA-Interim).

line 262:That wasn't clear to me. I thought the scaling factor was always used after being introduced in the method section. Not sure I have missed this while reading through the text, but I would suggest to make this clear while showing CE for the first time (if not done and I just missed that).

We decided to introduce the scaling factor only in section 3.3. In doing so, the reader will only be confronted with the scaling factor once we use it, and this hopefully will clarify this point.

lines 274-275, Fig. 10:I would suggest to include the 0-line in the anomaly plots, which makes the reading of those plots much easier. This would also highlight the issue of the blocking classifications, that this one does for some regions seems to be only able to capture negative anomalies (10b, d and h). Further, it seems a bit confusing when talking about amplitudes while interpreting this plot. In my mind I quickly switched to my usual interpretation of amplitude, representing the strength of the anomaly. This was confusing, because e.g.Fig. 10b and d seem to indicate that for the blocking reconstruction negative amplitudes of the anomalies are being overestimated while not being able to represent positive anomalies. But here amplitude refers to a from the CE calculation, right? I would suggest to mention this again explicitly, which makes following the interpretations easier.

We added the 0 line to Figure 10. We also changed the formulation in the text. Figure 10 shows the reconstructed magnitude of precipitation and temperature, so the "usual" interpretation. It is not a = std(observed)/ std(predicted).

2.3 Discussion and Conclusion

line 286: Although, the better reconstruction is more focused towards western Europe, whereas temperature is better reconstructed at more eastern and southeastern Europe (Fig. 9). So what does in general mean, a higher mean CE value, occurrence of very high CE values or more points showing CE >0.25? I think the authors could present this a lot more convincingly, if they could attribute a quantification to it, using some sort of simple measure as mentioned above. Just by looking again at Fig. 9 this doesn't seem to be so obvious. I appreciate the authors give some reasoning behind the differences between temperature and precipitation in this paragraph, which represents some explanation beyond just showing the results of this correlation/CE test.

We thank the reviewer for this comment, which raises an interesting point of what constitutes a "better" reconstruction, which will depend on the specific question being asked. We wish to avoid presupposing the questions people might use such reconstruction methods for, and therefore putting our judgement on what counts as "better". Nevertheless, we added a statement in lines 373 and summarizes the general differences and findings in l. 374-385.

line 305: So what is the actual main message of the manuscript. As long as one chooses classifications which are associated with large scale flow anomalies (NAO, blocking, jet), one will be able to capture to some extent the seasonal large scale anomalies over Europe. But this study does not come to a conclusion what metric is best and it also depends on what someone might be interested in (precipitation or temperature). So the main message is that there is no perfect metric to capture both temperature and anomalies? A possible limitation of the reconstruction could be due to extreme events, but those are not considered in this study.

The review is correct, there is no perfect metric to capture both anomalies and the performance depends on the specific region we are interested in. We apologise if this message did not come across that clearly. We added a sentence (l.386-387) to the conclusions section to emphasise this point.

lines 298-304:I think I made this point already earlier: isn't this (necessity of including a scaling factor) also a problem of the reconstruction method? If one would only use NAO values above 2 or below -2, it would be obvious that the reconstruction would be underestimate the amplitude, wouldn't it? So going down with this threshold to 1 would probably do a better job in reconstructing the seasonal anomalies, because more seasons will be represented within this class and more variability is possible. Whereas defining a class of NAO with values above -5 would do a very bad job, because it will only reproduce climatology. So there is some sort of useful cutoff, or the classification are already defined in a useful way (by EOF, clustering, etc are they already constructed in a way to capture the variability). So I assume my question is, if this shouldn't be rather discussed with more focus on the method / discussion of how necessary a scaling is to start with? Further, I think it was not fully clear in the manuscript which definition of CE was used (I raised this point also in another comment). It wasn't clear to me that the first results were for the non-scaled CE values (sorry if I just missed it). But also the choice of CE (Equ. 4 or Equ. 6) was not fully clear to me. If I am not mistaken, Equ. (4) was taken to calculate the CE values, right? Was Equ. (6) also used somewhere? In general, what is the reason to use Equ. (4)? Since Equ. (6) doesn't really need more calculations or is too complicated, it is not obvious for me why exactly the simplification was used? Is the reason behind this the identification of a useful scaling (Fig. 8)? I think the authors could make this a bit clearer. Also, I hope I didn't miss it, but did the authors compare the two calculations, how different are the results? This information could be useful to justify the general use and handling of Equ. (4).

This is a very interesting comment. Actually, we do not think that it is obvious that using "NAO values above 2 or below -2" instead of 1 "would underestimate the amplitude" because both the frequency of occurrence but also the strength of the anomaly patterns are expected to change.

This comment is addressed by the additional regression analysis completed, discussed in section 4.1 (and Fig. 11). Please see a fuller answer in response to an earlier comment. In short: by performing the reconstructions using regression analysis on 2m temperature and the NAO we find that, when using an ordinary least squares regression, a very similar scaling factor is required as for the main composite analysis. As the regression includes all days, this suggests that it is not the exclusion of "undefined" days that is the root cause of the scaling factor. This conclusion is supported by the sensitivity analysis performed on the jet cluster analysis presented in section 1 in the Supplementary

material, showing that including all days produces a very minor reduction in the magnitude of the scaling factor, but does not remove the need for it.

For the definition of CE, we have now used the full equation (3) and not the simplified version (former eq 4). The simplification (former equation 4) was used in Figure 8 for calculating the scaling factor. The simplification works only if the mean of the reconstructed anomalies ($\overline{p}$) is close to zero. This is given for the NAO and the jet configuration, but not for the blocking (as shown now in Figure S2). Therefore, the calculation of the scaling factor has been performed only for the NAO and the jet. We have clarified this point in the manuscript (lines 288-291).

lines 305-314 and lines 315-324: Really interesting aspect in those two ara graphs. But the comparison between NAO (with large region) and jet (with smaller region) is difficult, because of different underlying variables. Further, a larger region does not necessarily help to get better correlations, as large part of the NAO region is also far way from Europe. Therefore, the argument of the NAO region has the advantage to capture also the European region is not really convincing, as the NAO, as mentioned by the authors, also only includes 2 classifications for the whole region. Apart from this, the jet-region also includes part of western Europe, and as Europe is part of the west wind regime, it is not really surprising that the upstream information has an impact on the immediate downstream weather response. Extending the jet-region eastward would be very interesting to see, if this indeed does substantially increase the reconstruction skill, but I would assume this is not clear. So this point is very interesting, but it was not investigated here. Same is true for the next interesting point (number of classifications for one variable), also very interesting but also not included in this manuscript. The question about the relevant or optimal number of classes for the reconstruction exercise does sound also really interesting. But it is not clear to me, what the example of the 30 MSL classifications tells the reader, as at some point maybe too many classifications could be reducing the skill in a similar way as extreme events could be problematic. The more classifications, the more relevant individual extremes could become, because their impact will be felt more strongly during individual seasons.

The main purpose of the study was to compare the ability of three classification methods (NAO, blocking, jet configurations) to reconstruct seasonal climate anomalies. While it is for sure interesting to investigate how reconstruction can be optimized (e.g. following the suggestion mentioned by the reviewer), such analyses are beyond the scope of this study. However, we can certainly address some of these issues in a revised manuscript.

With the example of 30 MSL, we wanted to discuss that retaining too much information (i.e. using 30 MSL and all days, which also includes quite some noise) does not necessarily lead to better skill than using only 2, 3 or 4 patterns. Therefore, too much information does not necessarily translate into better reconstruction skill. Similarly if we include all jet days (see sensitivity test) only small improvements are achieved. We reformulate this part (l. 326-335)

To illustrate why we feel the region selected for analysis could play a role, here is an example. The NAO region covers much more of Europe than the jet domain. Since the NAO is calculated by EOF analysis that reveals the pattern that maximises the variance in that region, it can be expected that using a region that extends eastwards of 0° will better capture the variability over Europe and Scandinavia. To illustrate this, we calculate the NAO index using daily SLP data for the NAO domain (1st plot) and the jet domain (2nd plot). The NAO pattern is visible from both analyses, but the patterns (EOFs) are not identical and we see that the influence of the NAO extends eastern of 0° (left plot). When looking at the temporal evolution of PC1, we see that even if the two time series are correlated, the days that will be classified as NAO+/- (exceeding the 0.5 standard deviation, 3rd plot) will not be exactly the same and could lead to different reconstructions.

[Figure]

lines 325-330:Same as for lines 305-314 and 315-324. The authors nicely present a way of how their results can be interpreted, where relevant impacts of the method on the results might occur or how specific changes in the classification setup could have an impact. I find this discussion very helpful, but I am somehow missing the part where the authors are highlighting the results of their study. Main issue with this point can be found in my general comment.

To address the reviewer's concern, we have now separated the discussion part from the conclusion to better highlight the implication of the study.

**Reviewer 2:**

The paper presents an analysis on the ability of different circulation patterns frameworks to describe the seasonal wintertime precipitation and temperature anomalies over Europe, using ERAInterim reanalysis data. Three different frameworks are used: the two NAO phases, blocking over three different regions and five Atlantic jet stream clusters.

The paper is clear and well written in all parts, and covers a current hole in the literature regarding circulation patterns, presenting original results which can be very useful for the community. I therefore recommend the article for publication in Weather and Climate Dynamics, once the following minor comments are considered.

General comments/suggestions

- I was very pleased to see the topic of this work, which I think was missing in the literature, and really enjoyed its reading. As a minor comment, I don't really understand the authors' choice not to consider - amongst the sets of circulation patterns - the "classical" four Euro-Atlantic weather regimes framework, which has been (and still is, to my knowledge) the most used in literature. I think it would be in the authors' interest to show the main results (e.g. Fig. 6 and 7/9) also for the 4 regimes framework, which would be of interest for many works on regimes when discussing the related impacts. I respect the authors' choice and I'm not asking to repeat the analysis for the k=4 case, but maybe some comments in the text and conclusions referring to the correspondence between the k=5 and k=4 regimes would help the reader "translating" the results to that framework.

We agree that the four Euro-Atlantic winter regimes widely used in the literature. We compared the different k in a previous study (Madonna et al 2017), using the jet latitude index (3 regimes), the classical weather regimes, and the jet cluster using k=3,4,5. Even if using different variables (zonal wind vs. geopotential height) and domains, in Madonna et al 2017 we showed the 4 jet clusters correspond well to the 4 weather regimes. Using 5 jet clusters, the analysis revealed that the zonal regime is a combination of central (48%) and tilted (37%) jet (their Figure 10). We included part of this

discussion in section 2.1.3. (lines 83-92) and also reference the correspondent weather regime when we present wind, precipitation, and temperature anomalies (lines 180-181 and 192-194).

- As discussed by the authors, one of the problems of the reconstruction is the underestimation of the seasonal anomalies, due to the limits of the mean composites. Maybe a simple estimate of how much does the variability impact the reconstruction in the different regions would be given by the ratio between the standard deviation and the mean of the composite patterns. Just a suggestion.

We thank the reviewer for the suggestion. We computed the ratio std(x)/mean(x) for the composite pattern (Supplementary Figure S7 and S8). We expected to have large values of standard deviation within each pattern, as noted previously in lines 249-25, and this can be a reason for the underestimation of the amplitude of the reconstruction. We also perform an additional analysis using regression that seems to confirm that the underestimation is linked to the variability within each composite (see lines 345-363 for the entire discussion).

Specific comments

- L35. "..the North Atlantic jet stream can assume five different configurations..": the number of clusters to be considered is a matter of debate in literature and probably will not lead to a conclusive "best number", since all choices retain some level of arbitrariness. Please acknowledge here that this is the authors' choice, while other choices for the number of regimes are possible (e.g. as in Madonna, 2017).

We agree with the reviewer and added a comment to acknowledge that (lines 86-92).

- L70. Please briefly comment on the choice of 10%. Have you tried with other thresholds? Are the results sensitive to this choice?

The choice of 10% (i.e. 0.1) is to a certain extent arbitrary and arose as we did not want to select days where only a few gridpoints were blocked. The Figure on the left shows a histogram with the fraction of gridpoints (x-axis) in the "Greenland box" for all days with blocking, while the Figure on the right shows the number of blocked grid points for a subset of 120 days. The 10% (i.e. 0.1) threshold is marked by the red line. From the time series (right), we see that most of the points with values below 0.1 are followed by points with a larger threshold, which means that the block is building and the 0.1 threshold captures it only at a later stage. Translated to the occurence of blocked days per season, this means that the absolute frequency depends on the threshold, but not the number of blocking events. We have included a comment in lines 76-78.

[Figure]

- L83. How is *d* calculated? Please show the formula here or add a reference.

d is the normalized inverse Euclidean distance from the centroid. The Euclidean Distance (E) for a point x from the centroid in a n-dimensional space is defined as $E = \sqrt{\sum_{i=1}^{n}(centroid_i - x_i)^2}$ . d is then the inverse of the normalised distance, d=1/Enorm, where the normalisation is Enorm = E /max(E). This information has been added (l. 99-100).

- L86. Is this method and the threshold of 0.5 used elsewhere? If so, please add a reference here. Instead please briefly discuss the choice and its possible impacts on the results.

The choice of d=0.5 is somehow arbitrary but was introduced to be sure to capture only days that belong to the given centroid. We have now provided a sensitivity test to discuss the possible impact of this choice in the supplement.

- L105. How close to a zero anomaly are the undefined days? I imagine the mean is very close to zero, but this may also come from cancellation of opposite anomalies. I'd suggest to show the standard deviation of the anomalies for the unclassified days (in the supplementary), to assess whether their exclusion might impact the skill of the reconstruction. The underestimation of the amplitude of the seasonal anomalies might also be linked to the filtering, since the denominator in equation 2 is always 90, but the number of assigned days is usually much less.

We added Figure S1 to show that the anomalies of the undefined days are close zero. We also added the standard deviation for the composites of all categories in the supplement (Fig. S7 and 8). The standard deviations in some regions are as large as the mean values. This can be understood from a synoptic perspective: for example, if two storms follow a similar path, the exact location where the most intense precipitation falls can be quite different.

We also performed a sensitivity test for the jet cluster using all days to reconstruct the seasonal anomalies (see Supplement Section A). Including all days reduced only marginally the underestimation of the amplitude (the scaling factor β). We also performed an additional analysis with a regression approach to better understand why we need the scaling factor β (see paragraph in the discussion, lines 345-363 ). The analysis suggests that the scaling factor is needed because of the large variability within each classification pattern and not because of the filtering..

- L129. CE>0.25. Is this threshold used elsewhere? If so, please add a reference.

We discuss the choice of CE>0.25 using a few examples (also linked to the suggestion of reviewer 1).

l. 152-161: "When applied to reconstructions and observations, the CE is a measure of skill in reconstruction that is more restrictive than a simple correlation because it penalizes for both phase and amplitude misfits. For a perfect reconstruction, CE=1. For a reconstruction with observed variance (a=1) that is correlated with the observed time series of seasonal anomalies at r=0.5, the result is CE=0. For a reconstruction that is perfectly correlated with observations but with twice the observed amplitude, we arrive at the same result of CE=0. We consider CE>0.25 to indicate a good reconstruction. A reconstruction that has perfect variance (a=1) and CE= 0.25 would explain 39% of the observed variance (r=0.63); given 35 years of data (degrees of freedom), this correlation would be significant at p=0.0001. For a reconstruction with perfect correlation (i.e. r=1), CE>0.25 can be obtained for amplitude values 0.13< a <1.87. As the CE maximizes when r = a (and max(CE)= r 2 ), a CE of 0.25 also implies a minimum correlation of 0.5 independent of the amplitude error. For this reason, only points with correlation greater than 0.5 are considered when optimizing the CE (see Sec. 7)."

- L135. Is the "/" a typo?

Thanks.

- Section 2.2.2. I found it quite hard the reading of the section before reading Section 3.3 and looking at Figure 8. To make this easier, I suggest to explicitly refer to Section 3.3 and Figure 8 in the text. Also, a possibility would be to move Section 2.2.2 to the beginning of Section 3.3, since the scaling factors are not used till then.

We agree that the introduction of the scaling factor in the method, but their application only in 3.3. makes the reading and understanding of this part hard. We moved its introduction at the beginning of section 3.3, as suggested by the reviewer, to improve the clarity of the manuscript.

- L138. "small". with respect to the mean?

With respect to standard normal distribution (i.e. μ=0, σ=1), but since this might be rather confusing, we removed it.

- L163. "are placed to allow easy comparison": has some automatic matching of the wind patterns been performed (e.g. maximizing the pattern correlation)?

In Madonna et al. 2017 the wind classification has been explicitly compared with blocking and NAO. There has been a temporal match (i.e. by looking at the number of overlapping days in each definition). A spatial match has been performed only between the different jet clusters in Madonna et al 2017 (i.e. for k=3,4 and 5, their Table 1). A footnote has been added.

- L174. The S-jet cluster is made up by less days, so that would probably enlarge the anomalies.

Let's assume that the S-jet is a subsample of the NAO-. If the S-jet days are equally distributed within the NAO- days, the anomalies for S-jet should not be enlarged just because we use less days. But the anomalies would be enlarged if the subsample of S-jet belongs to the upper tail of the NAO- distribution, or reduced if they are located in the lower tail. Therefore, we do not expect the anomalies to be stronger just because fewer days are used.

- L178-179, L197. In the comparison with NAO+, the usual 4 regimes would look more natural to me. Probably a sum of the tilted and central jet states would show similar anomalies, as it is said in the text, and also correlate better with the NAO+ timeseries. It might be worth adding a comment on this in the text.

We added a comment in the text (lines 180-181 and 192-194) and also added a more general comparison between jet and weather regimes in section 2.1.3. with reference to the discussion to Madonna et al. 2017. In their study, the 4 classical weather regimes and the jet clusters (k=4 and 5) have been directly compared. In fact, the NAO+ regime (i.e. the zonal regime) is a mix of central (48%) and tilted (37%) jet (their Figure 10).

- L200. The unblocked days are different from the undefined/neutral days, so I won't put them in the same category. I expect (at least part of) them to correlate with positive NAO and have similar anomalies. I suggest to add a comment on this in the text.

We agree that the unblocked day category includes days that belong to the "zonal flow/NAO+ regime" and days that are undefined. This choice was intentional, as the goal was to assess how much blocking episodes influence the seasonal signal. We added a comment on that in the manuscript (l 198-203).

- L201. See comment at line 105. I think it would be interesting to show the mean (even if close to zero) and standard deviation for these days in the additional material.

We added these Figures in the supplement (Fig. S1).

- Section 3.2. Have the reconstructions been performed for the wind also? Table 2 and L260 seem to suggest so. Why are the results not shown? It might be worth commenting on this briefly in the text.

Yes, we did that for the wind as well. We presented only precipitation and temperature on the one hand because as seasonal means they have a stronger surface impact than the wind, and on the other hand because the winds/circulation is somehow included in the definition of the categories. We added the correlation and CE for wind in the supplement (Figure S3 and S5) and added a brief comment in the text (l. 233, 263-265).

- L216. "that" -> than?

Thank you

- L224. Also over France the blocking method has substantially better skill.

In the original version, we used the simplification of the calculation of the CE, which is a good approximation if the mean of the reconstructed anomalies is close to zero. While this was true for the NAO and the blocking, the reconstructed anomalies for blocking were not zero. We added this information in the manuscript (288-291) and in the supplement (Fig. S2) and have modified the text accordingly. The comment for France has been added (l.322-324 and 382).

- L227-8. "...regions with poor correlation skill mostly have low temperature variability..". Is this referred to the NAO phases or to all methods? I don't think this is true in general. For example Spain and southern Italy show low temperature variability but good skill for blocking/jet regimes.

The reviewer is right and this part was not clear and not valid for all methods. We reformulated this sentence (l 243-244).

- L240. A large skill is also apparent over North Africa/the Mediterranean, I'd add a comment on this in the text. Also, if the figures are already available, it might be worth showing the equivalent of Figure S3 for the correlation also.

We added the correlation for a larger domain to the supplement (Fig S3) and a comment regarding North Africa in the text  (l263 ).

- L292-297. I'd comment on the case of France as well, which is well represented only by the blocking framework.

We have rewritten and rearranged most of the paragraph, but we also have included a comment for blocking over France (l. 382).

- L302-304. Also, this might be related to the filtering, see comment at line 105.

We performed the same analysis using all days (see sensitivity test in the Supplementary material), and including all days did not reduce the scaling factor. To understand the need for the scaling factor, we performed for the NAO a supplementary analysis, regressing the NAO index on daily anomalies. The results are presented in the discussion and suggest that the filtering is not the main cause for the underestimation of the anomalies, but rather  the large variability within each classification.

- L311. The "classical" 4 Euro-Atlantic weather regimes are usually calculated using larger domains, extending up to 40 degrees east. Do you think this could increase the skill in central/eastern Europe?

If much of the variability over easter Europe is captured by the first 5 EOFs, that could be the case. We know that the circulation over the North Atlantic plays an important role for the European region

and that there is a good correspondence between the jets (k=4) and the weather regimes (from Madonna et al 2017). Using a metric that covers a larger domain might capture more of the variance over eastern Europe and increase the reconstruction skill. However, we have not tested that yet, but could be a nice additional analysis.

---

## Referee Report (RR1)

I thank the authors for their thorough reply to all my raised points of the previous revision process. As the authors implemented all my previous suggestions and/or added additional text/discussion to clarify my previously raised points, I am happy to see the current version of the manuscript published. Below are some final minor comments that may be addressed.

**lines 101-102:**
I appreciate the inclusion of this extensive sensitivity analysis. Strong variations in the choice of d do not seem to have significant impact on the results/conclusion. I would suggest to explicitly mention this robustness of the results here, because only refering to the sensity analysis in the supplement could lead to the impression that the conclusion might not be that clear.

**line 151:**
Inconsistent use of this abbreviation, sometimes Sec. and sometimes section (e.g. line 256), same for Fig. (Fig. and Figure).

**line 215:**
"...by almost no anomalies (Fig. S1)." I would suggest to rephrase this slightly. The main point is clear here, but it is maybe a bit too much to talk about almost no anomalies, as the temperature anomalies of Fig. S1 (e.g. high latitudes) are close to the anomalies discussed previous (e.g. Fig. 4, cold anomalies in Africa for N-jet). Of course, for this example I compared the anomalies for Africa with the ones for high latitudes and not the anomalies limited to the same regions. Maybe the sentence could be rephrased in a way to account for this.

**lines 351-352:**
weighted orthogonal is written in another format than the rest of the text.

---

## Author Response (AR2)

**Second Revision of wcd-2021-6 "Reconstructing winter climate anomalies in the Euro-Atlantic sector using circulation patterns" by Madonna et al.**

We would like to thank the reviewer for the comments and suggestions.

I thank the authors for their thorough reply to all my raised points of the previous revision process. As the authors implemented all my previous suggestions and/or added additional text/discussion to clarify my previously raised points, I am happy to see the current version of the manuscript published. Below are some final minor comments that may be addressed.

lines 101-102:

I appreciate the inclusion of this extensive sensitivity analysis. Strong variations in the choice of d do not seem to have a significant impact on the results/conclusion. I would suggest to explicitly mention this robustness of the results here, because only referring to the sensitivity analysis in the supplement could lead to the impression that the conclusion might not be that clear.

We replaced "A sensitivity test to the choice of this threshold is provided in the supplement" with "A sensitivity analysis suggests that the choice of the threshold d does not have a significant impact on the results (see supplement)."

line 151:

Inconsistent use of this abbreviation, sometimes Sec. and sometimes section (e.g. line 256), same for Fig. (Fig. and Figure).

We now use Figure throughout the manuscript.

line 215:

"...by almost no anomalies (Fig. S1)." I would suggest to rephrase this slightly. The main point is clear here, but it is maybe a bit too much to talk about almost no anomalies, as the temperature anomalies of Fig. S1 (e.g. high latitudes) are close to the anomalies discussed previous (e.g. Fig. 4, cold anomalies in Africa for N-jet). Of course, for this example I compared the anomalies for Africa with the ones for high latitudes and not the anomalies limited to the same regions. Maybe the sentence could be rephrased in a way to account for this.

We modified lines 215. It now reads "The composites of precipitation and temperature for those categories are similar to climatology and therefore their patterns are characterized by little anomalies, in particular over the European continent (Figure S1)"

lines 351-352:

weighted orthogonal is written in another format than the rest of the text.

We have removed the italic notation.